# Integrative computational, synthetic, experimental evaluation of targeted inhibitors against matrix metalloproteinase-9: Toward precision modulation of proteolytic activity

Zainab Ahmed Rashid[1], Dima A. Sabbah ●[2]*, Kamal Sweidan[3], Rima Hajjo[2,4], Sanaa K. Bardaweel ●[1]*

**1** Department of Pharmaceutical Sciences, School of Pharmacy, University of Jordan, Amman, Jordan, **2** Department of Pharmacy, Faculty of Pharmacy, Al-Zaytoonah University of Jordan, Amman, Jordan, **3** Department of Chemistry, The University of Jordan, Amman, Jordan, **4** Laboratory for Molecular Modeling, Division of Chemical Biology and Medicinal Chemistry, Eshelman School of Pharmacy, The University of North Carolina at Chapel Hill, Chapel Hill, North Carolina, United States of America

* dima.sabbah@zuj.edu.jo (DAS); s.bardaweel@ju.edu.jo (SKB)

## Abstract

Matrix metalloproteinase-9 (MMP-9) is a zinc-dependent enzyme that degrades the extracellular matrix and is involved in various diseases, including rheumatoid arthritis, atherosclerosis, tumor invasion, and metastasis. Despite the development of inhibitors, none have succeeded in trials. Our goal was to find potential inhibitors to regulate its proteolytic activity. Ligand- and structure-based drug design approaches were explored to identify inhibitors against wild-type (1GKC) and mutant (2OW1) MMP-9. A pharmacophore model was created, and drug-like molecules were prioritized to guide the development of benzamide and 1H-indole-2-carboxamide derivatives. These compounds were synthesized and characterized using $^{1}$H NMR, $^{13}$C NMR, and HRMS (ESI). An experimental evaluation assessed their inhibitory potential and IC$_{50}$ values against MMP-9. Most tested inhibitors fit the pharmacophore model, which consists of three aromatic/hydrophobic spheres and two hydrogen-bond donors/acceptors. Compounds **1**, **2**, **8**, **10**, **20**, **21**, **27**, and **29** exhibited significant inhibition (P < 0.0001) of over 60%. Compounds **2** and **20** inhibited growth by over 70%, with IC$_{50}$ values of 28.59 µM and 30.82 µM, respectively. The IF docking showed strong binding for these, with scores of −9.179 and −10.739 kcal/mol. The alignment between the computational approach and experimental validation reinforces the inhibitor's specificity and potency, confirms the docking model, and suggests that the predicted binding pose represents key biological interactions.

**Data availability statement:** All relevant data are within the manuscript and its Supporting Information files.

**Funding:** The author(s) received no specific funding for this work.

**Competing interests:** NO authors have competing interests.

## Introduction

Matrix metalloproteinases (MMPs), zinc-dependent neutral endopeptidases, can degrade all components of the extracellular matrix [1]. Based on their specific structural features, they are classified as collagenases, gelatinases, stromelysins, and matrilysin [2]. Among all, matrix metalloproteinase-9 (MMP-9) is involved in several pathological processes such as rheumatoid arthritis, atherosclerosis, tumor growth, invasion, and metastasis [3–7].

The structure of MMP-9 includes a signal peptide, pro-peptide, catalytic domain with a highly conserved zinc-binding site, fibronectin domain, and a hemopexin-like domain, which are connected via a hinge region [8]. The catalytic domain of human MMP9 contains a five-stranded β-sheet and three α-helices, similar to other MMPs [1]. Various structural features identified in the MMP-9 catalytic domain are conserved across MMPs [1]. Regarding these features: 1) an S-shaped double loop formed between strands β3 and β4, which clamps both the structural $Zn^{+2}$ and one $Ca^{+2}$, 2) a "bulge-edge' involving residues at the N-terminal and part of strand β4 that is essential for inhibitor binding, 3) a conserved sequence motif HExxHxxGxxH characteristic of the metzincin zinc-endopeptidase superfamily, 4) conserved residues Pro 421-X-Tyr 423 forming the wall of the S1 Table` pocket, and 5) the specificity loop (Arg 424-Pro 430) curling behind the S1 Table` pocket. In the MMP-9 structure, two $Zn^{+2}$ ions are observed—one catalytic and one structurally essential. The catalytic zinc is located in the active site, forming tetrahedral bonds with three histidine side chains, including His 401, His 405, and His 411, which are part of the consensus sequence (HExxHxxGxxH) that shapes one end of the active site cleft. The fourth bond is with the Cys 99 side chain of the pro-peptide domain. On one side of the active site, His 401 and His 405 residues extend from adjacent turns of helix αB into the active site cleft. The MMP-9 protein curves at the end of the cleft, forming a sharp turn back toward the catalytic $Zn^{+2}$, positioning His 411 as the third ligand.

Most MMP-9 inhibitors are classified based on their $Zn^{+2}$ binding groups. The most common group is hydroxamic acid, which binds to a catalytic zinc ion via two identical bonds and extends into the S1 Table' selectivity pockets. They exhibit nanomolar potency against MMPs. The lack of selectivity, poor pharmacokinetics, and limited oral bioavailability of these inhibitors have led to their failure in clinical trials. However, other studies have reported the selectivity and potency of various other $Zn^{+2}$ binding groups, such as hydroxamates, reverse hydroxamates, carboxylates, thiols, and phosphinates [9,10]. Other findings suggested that high-affinity $Zn^{+2}$ binding groups can serve as a foundation for further modifications to develop new patterns for selective inhibitors. [9]. Several synthetic MMP-9 inhibitors have been tested in phase III trials and have shown no therapeutic efficacy [10]. Many factors have contributed to the failure of MMP-9 inhibitors, including high toxicity that causes severe musculoskeletal side effects and a lack of specificity, which then results in unexpected side effects [11,12]. Recent advances in computational methods for prioritizing drug-like molecules (hit compounds) and creating highly selective inhibitors for other MMPs suggest that targeting metzincins specifically might be possible [13,14].

Interestingly, MMPs are used across various research fields, including biochemistry, cell biology, pathology, immunology, bioinformatics, and computational biology. From this perspective, our proposed multi-objective approach aims to develop a new ligand-based pharmacophore model based on key chemical features of selective MMP-9 inhibitors, screen the pharmacophore against the NCI database using ligand- and structure-based methods to identify potential hits with novel scaffolds for MMP-9, purchase the commercially available hits, synthesize analogs of these hits, and evaluate the inhibitory potential of the candidate MMP-9 inhibitors through molecular docking and experimental validation evaluation.

## Materials and methods

### Computational methods

**Pharmacophore generation.** The 3D structures of all reported selective MMP-9 inhibitors (at least five-fold selectivity) retrieved from different databases were drawn and energy minimized using the MMFF94X force field in the Molecular Operating Environment (MOE) [15] software based on the template of the co-crystallized ligand (7MR) in MMP-9 (PDB ID: 2OW1) [16]. Each minimized structure was aligned with the 7MR structure. Then, a flexible alignment was carried out using an energy cutoff value of 15. The structures that aligned perfectly were used to create the pharmacophore model with the Pharmacophore Query model in MOE; the threshold was set to 50%, and the tolerance was set to 1.6.

### Pharmacophore search

Ligand- and structure-based approaches were used to identify potential hits. The NCI database was first downloaded from the NCI Cheminformatics Tools and Services website [17]. Before the hit searching process, this data was processed based on molecular weight (m.wt) (g/mol), excluding all structures with m.wt < 200 and >600.

Two hit-search methods were used in the ligand-based approach: the first involved partial matching of four out of five pharmacophoric features against the treated NCI database (214,535 structures), while the second involved searching for an exact match of all five features in the same database. The hits were then filtered using the Lipinski rule of five (excluding molecular weight) to identify drug-like molecules. These optimization steps included criteria such as logP (octanol/water) ≤ 5, hydrogen bond donor (HBD) ≤ 5, and hydrogen bond acceptor (HBA) ≤ 10.

### Hits

**Compound 1** and **Compound 2** represent the two purchased hits from AmBeed (USA) (Fig 1). Both are partial match hits with developed pharmacophore (fit four pharmacophoric features), which found available commercially and not yet been reported in literature as MMP-9 inhibitors.

### Preparation of protein structure

The X-ray crystallographic structures of the wild-type (WT) and mutant-type (MUT) MMP-9 (PDB IDs: 1GKC and 2OW1), with resolutions of 2.30 A° and 2.20 A°, respectively, were retrieved from the RCSB Protein Data Bank [16,18]. Then, we adopted the assembly of 1GKC and 2OW1 and initiated protein preparation and refinement protocols using the MAESTRO enterprise [19]. The protein structures were pre-processed to assign bond orders, add H, create zero-order bonds to metals, form disulfide bonds, convert selenomethionines to methionines, fill missing side chains using the Prime algorithm, fill missing loops with the Prime algorithm, cap termini, delete waters beyond 5 A°, and generate het states using Epk: pH 7 ± 2. Then, the proteins were refined and optimized for H-bond assignment. This process involves automatically optimizing all the H-bonds. Additionally, interactive optimization allows for the effective adjustment of various clusters of H-bonded species. Further refinement of the protein was performed by minimizing the side chains while restraining the protein backbone amino acids, using the optimized potentials for liquid simulations-2005 (OPLS_2005) force field.

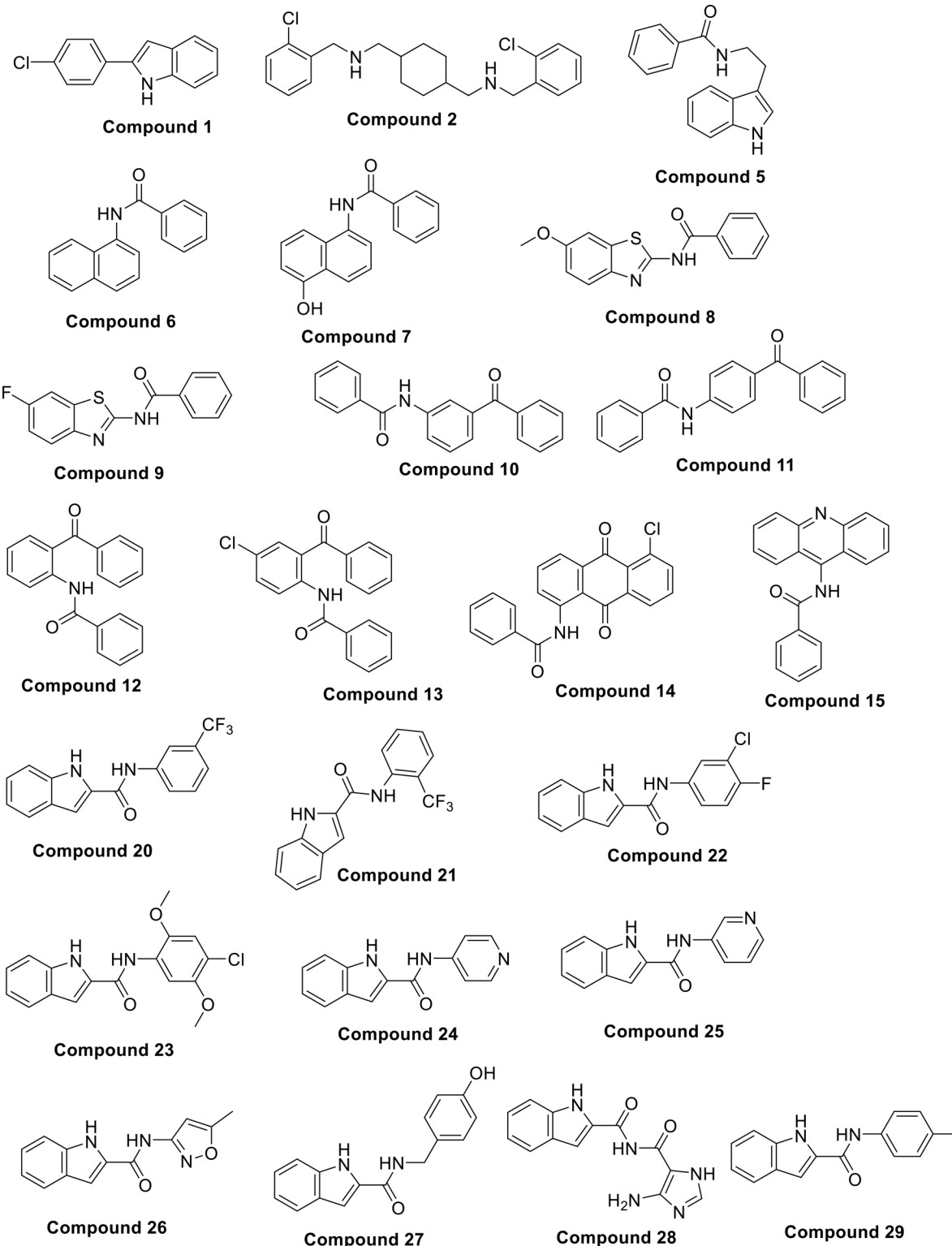

**Fig 1. The blue border indicates the chemical structures of the retrieved hits obtained through ligand- and structure-based drug design approaches.** The green border indicates the chemical structures of benzamide derivatives. The orange border indicates the chemical structures of 1H-indole-2-carboxamide derivatives.

## Ligand preparation

The ligand preparation (LigPrep) algorithm in MAESTRO software is a powerful set of tools designed to generate high-quality 3D structures of numerous verified MMP-9 inhibitors (ligands) that are modeled on the co-crystallized ligand (7MR) template in 2OW1 [19]. The tautomers and ionization states were generated. The force field geometry was optimized using the MacroModel wizard with the OPLS_2005 force field.

## Glide grid-generation

The receptor grid box was generated using the Glide Grid generation script [20]. This step is crucial before ligand docking, which requires a pre-optimized protein structure with accurate bond orders and formal charges. The grid generation panel features three tabs for generating a grid. These tabs include receptors, sites, constraints, and ratable bonds.

## Ligand-based virtual screening

A virtual screening process was used to filter potential hits based on their binding affinity scores, expressed as docking scores (Kcal/mol), and to predict their binding interaction mode within the MMP-9 crystal structure (PDB ID: 1GKC) [18] using the script ligand-based virtual screening in MAESTRO [19]. Before conducting the virtual screening, three main steps were carried out: hits preparation, protein preparation, and grid generation as previously described. In the input tab, we specify the LigPrep file, which contains the pre-prepared hits. The receptor binding site must be in a Glide-grid file to facilitate the screening process.

## Glide docking

The ligand docking panel was used to perform the flexible ligand docking process. Glide enables flexible ligand docking into rigid protein structures, accurately predicting the best ligand poses and ranking ligands based on their predicted affinities. The pre-calculated protein grid was specified, and extra precision docking was selected. Within the ligand docking panel, we specified the pre-prepared ligands file to be docked and assigned it directly. Then, we set the ligand van der Waals scaling for nonpolar ligand atoms. Ligand atoms were scaled based on their absolute partial atomic charges less than 0.15, with a default scaling factor of 0.80. All other parameters were set to their default values. The binding affinity was expressed as a Glide score (kcal/mol). The more negative the Glide score, the better the predicted ligand-protein binding mode.

## Induced fit docking (IFD)

Initially, the optimized and energetically prepared protein structure was imported into the workspace. We selected the Lig-Prep file of the prepared ligands for use in docking. The receptor was distinguished from the complexed ligand to create a Glide grid accurately. In the receptor tab, the box center option was selected to identify the centroid of the workspace protein. This preliminary docking was performed with a softened receptor and ligand by scaling their van der Waals radii. By default, the scaling factors for both the receptor and ligand were set to 0.50. We specified the refined residue within 5Å of the co-crystallized ligand position and selected the side chain optimization option. The remaining options in this tab were left at their default values.

## Experimental

**Chemistry.** All chemicals and solvents were supplied as follows: dimethyl sulfoxide (DMSO), *N, N*-dimethylformamide (DMF), and n-hexane (95%) (M-Tedia). Chloroform ($CHCl_3$) (Emsure-Company), ethyl acetate (EtOAc) (Fisher-Scientific), acetone 99.8% (LABCHEM), and pyridine (BIOCHEM). 1-Amino-5-chloro-anthraquinone (Alfa-Aesar), Germany, 6-chloro-benzothiazol-2-ylamine, *p*-toluidine, 1-amino-4-hydroxy-anthraquinone, 1-amino-5-chloro-anthraquinone 2-amino-6-methoxybenzothiazole, 6-fluoro-benzothiazol-2-ylamine, 9-aminoacridine, 2-(1H-Indol-3-yl)-ethylamine (ACROS-Organics),

USA, oxalyl chloride 98%, 2-aminobenzophenone, 3-amino-benzophenone, 4-amino benzophenone, 2-amino-5-chloro-benzophenone, 4-aminopyridine, 3-aminopyridine, 5-amino-1-naphthol, 3-amino-5-methyl isoxazole, naphthalen-1-ylamine, 3-(trifluoromethyl) aniline, 2-(trifluoromethyl) aniline, 3-chloro-4-fluoroaniline, 4-chloro-2,5-dimethoxyaniline, 4-hydroxybenzyl amine, and 5-amino-4-imidazolecarboxamide (Sigma-Aldrich), Germany. All chemicals were of analytical grade and highly purified; hence, they were utilized without further purification. The hot plate and magnetic stirrer were purchased from Thermo Scientific Cimarec™. Rota vapor model R-215 (Buchi, Switzerland) was used to evaporate the ordinary solvents connected to the vacuum pump v-700 with a heating water bath (Hei-VAP value digital, Heidolph). Mp measurements of all the synthesized compounds were performed on the Gallenkamp Mp apparatus. $^1$H and $^{13}$C nuclear magnetic resonance (NMR) spectra were analyzed on Bruker-Avance III 500 MHz spectrometers with DMSO-d6 as solvent (The University of Jordan), coupling constant (*J*) values are given in Hertz (Hz), and chemical shifts are expressed in δ (ppm) using TMS internal reference. Thin-layer chromatography (TLC) was performed on 20×20 cm, 0.20 mm thick aluminum sheets pre-coated with fluorescent silica gel (Macherey-Nagel, Germany) and visualized under UV light (254/366 nm). High-resolution mass spectra (HRMS) were recorded using a Bruker APEX-IV (7 Tesla) instrument (The University of Jordan). External calibration was executed using an arginine cluster at a mass range of *m/z* 175–871, and the samples were dissolved in methanol and drops of formic acid.

## Synthesis of target compounds

Based on reported chemical procedures [21–24], two diverse chemical scaffolds with favorable properties and binding affinities, including benzamide and 1H-indole-2-carboxamide derivatives, were synthesized and characterized using $^1$H-NMR, $^{13}$C-NMR, DEPT-135, and HRMS (ESI). A supplementary file provided all relevant data on spectroscopic charts of the synthesized compounds.

## General synthesis of the benzamide scaffold

In a single-necked round-bottom flask attached to a reflux condenser and connected to a guard tube filled with magnesium sulfate (MgSO$_4$), benzoyl chloride (0.5 g, 3.5 mmol) in an anhydrous CHCl$_3$ (20 mL) was inserted to the reaction flask and stirred in an ice bath with an anhydrous pyridine (1.69 mL, 2.1 mmol, 6 eq). Then, a corresponding amine derivative (**4i-xi**) was dissolved in an anhydrous CHCl$_3$ (15 mL) and added portion-wise to the reaction in an equimolar (1:1) ratio relative to the nucleus. The solution was refluxed for 24 hr. At the end of the reaction time, the solvent was removed, and the residue was treated with CHCl$_3$ before being evaporated under reduced pressure. The residue was redissolved in CHCl$_3$ and extracted with distilled water to remove excess pyridine. The product was collected after solvent evaporation, followed by recrystallization with CHCl$_3$/n-hexane [21,22].

- ***N*-(2-(1H-indol-3-yl) ethyl) benzamide** (**5**)

Yellow powder; yield (72%); Mp 135−137°C; mobile phase (EtOAc: n-hexane) (1:3); Rf = 0.66; **$^1$H-NMR (500 MHz, DMSO-d$_6$)** δ (ppm): 2.97 (t, *J* = 7.6 Hz, 2H, H4), 3.58 (m, 2H, H3), 6.99 (t, *J* = 7.4 Hz, 1H, Ar-H4"), 7.07 (t, *J* = 7.5 Hz, 1H, Ar-H5`), 7.19 (d, *J* = 2.4 Hz, 1H, Ar-H2'), 7.35 (d, *J* = 8.1 Hz, 1H, Ar-H7'), 7.46 (t, *J* = 7.4 Hz, 2H, Ar-H3"/ H 5"), 7.52 (t, *J* = 7.2 Hz, 1H, Ar-H6`), 7.59 (d, *J* = 7.8 Hz, 1H, Ar-H4`), 7.85 (d, 2H, Ar-H2"/ H 6"), 8.61 (t, *J* = 5.8 Hz, 1H, NH amide), 10.81 (s, 1H, NH indole) ppm. **$^{13}$C-NMR (125 MHz, DMSO-d$_6$)** δ (ppm): 25.2 (C4), 40.3 (C3), 111.4 (C7'), 111.9 (C3'), 118.3 (C6'), 118.3 (C4'), 121.0 (C5'), 122.6 (C2'), 127.2 (C3"/C5"), 127.3 (C2"/C6"), 128.3 (C4"), 131.1 (C3a`), 134.7 (C1"), 136.3 (C7'a), 166.2 (C1) ppm. **Supplementary data charts (S#1)** in S1 File show the $^1$H-NMR, $^{13}$C-NMR, DEPT-135, and HRMS (ESI) spectrum of **compound 5**.

- ***N*-(naphthalen-1-yl) benzamide** (**6**)

Purple powder; yield (76%); Mp 139−142°C; mobile phase (EtOAc: n-hexane) (1:3); Rf = 0.45; **$^1$H-NMR (500 MHz, CDCl$_3$)** δ (ppm): 7.48–7.59 (m, 5H, Ar-H4'+ H5'+ H6'+ H7'+ H8'), 7.59–7.62 (m, 2H, Ar-H2'+ H3'), 7.76 (m, 1H, Ar-H5), 7.91 (m, 2H, Ar-H4 + H6), 8.00 (d, *J* = 7.8 Hz, 2H, Ar-H3 + H7) ppm. **$^{13}$C-NMR (125 MHz, CDCl$_3$)** δ (ppm): 108.8 (C2'), 110.2 (C4'),

113.5 (C8'), 114.8 (C8'a), 118.3 (C4/ C6), 119.1 (C7'), 120.5 (C3'), 125.2 (C4'a), 125.5 (C5'), 126.1 (C6'), 130.9 (C3/C7), 135.9 (C2), 143.3 (C1'), 145.4 (C5), 169.3 (C1) ppm. **Supplementary data charts (S# 2) in** S1 File show the $^1$H-NMR and $^{13}$C-NMR spectrum of **compound 6**.

- ***N*-(5-hydroxynaphthalen-1-yl) benzamide (7)**

Fade purple powder; yield (81%); Mp 265−267°C, mobile phase (EtOAc: n-hexane) (1:3); Rf = 0.23; **$^1$H-NMR (500 MHz, DMSO-d$_6$)** δ (ppm): 6.92 (d, *J* = 7.4 Hz, 1H, Ar-H6"), 7.33 (t, *J* = 8.0 Hz, 1H, Ar-H7"), 7.43 (d, *J* = 8.5 Hz, 1H, Ar-H2"), 7.47 (t, *J* = 7.8 Hz, 1H, Ar-H5), 7.54–7.60 (m, 3H, Ar-H8" + H4/H6), 7.61 (d, *J* = 7.2 Hz, 1H,Ar-H4"), 8.10 (t, *J* = 7.9 Hz, 3H, Ar-H3" + H3/H7), 10.21 (s, 1H, OH), 10.34 (s, 1H, NH amide) ppm. **$^{13}$C-NMR (125 MHz, DMSO-d$_6$)** δ (ppm): 108.6 (C2"), 114.3 (C6"), 120.9 (C4"), 124.5 (C8"), 124.8 (C8a`), 125.9 (C4a`), 126.9 (C7"), 128.2 (C4/ C6), 128.9 (C3/C7), 131.3 (C3"), 132.1 (C5), 134.0 (C2), 135.0 (C1"), 153.9 (C5"), 166.5 (C1) ppm. **HRMS (ESI)** m/z: Calculated for $C_{17}H_{13}N_1Na_1O_2$ (M + Na) $^+$ = 263.09463; found 286.08385. **Supplementary data charts (S# 3) in** S1 File **show** $^1$H-NMR, $^{13}$C-NMR, DEPT-135, and HRMS (ESI) spectrum of **compound 7**.

- ***N*-(6-methoxybenzo(d)thiazol-2-yl) benzamide (8)**

Yellow powder; yield (72%); Mp 135–138°C, mobile phase (EtOAc: n-hexane) (1:3); Rf = 0.32; **$^1$H-NMR (500 MHz, DMSO-d$_6$)** δ (ppm): 3.83 (s, 3H, OCH$_3$), 7.50 (d, *J* = 7.6 Hz, 2H, Ar-H4'/ H6'), 7.56 (d, *J* = 7.6 Hz, 2H, Ar-H3'/ H7'), 7.61–7.62 (m, 1H, Ar-H5'), 7.67 (s, 1H, Ar-H7), 7.95 (d, *J* = 7.7 Hz, 1H, Ar-H5), 8.13 (d, *J* = 7.7 Hz, 1H, Ar-H4) ppm. **$^{13}$C-NMR (125 MHz, DMSO-d$_6$)** δ (ppm): 55.5 (OCH$_3$), 104.5 (C7), 114.9 (C5), 117.9 (C4), 128.1 (C5'), 128.4 (C2'), 128.5 (C4'/ C6'), 129.1 (C3'/C7'), 130.6 (C6), 131.7 (C7a), 146.7 (C3a), 156.1 (C1'), 167.1 (C2) ppm. **HRMS (ESI)** m/z: Calculated for $C_{15}H_{12}N_2O_2S$ (M + Na) $^+$ = 284.06211 found 307.05188. **Supplementary data charts (S# 4) in** S1 File show $^1$H-NMR, $^{13}$C-NMR, and HRMS (ESI) spectra of **compound 8**.

- ***N*-(6-fluorobenzo(d)thiazol-2-yl) benzamide (9)**

White powder; yield (83%); Mp 138−141°C; mobile phase (EtOAc: n-hexane) (1:3); Rf = 0.3; **$^1$H-NMR (500 MHz, DMSO-d$_6$)** δ (ppm): 7.02 (td, *J* = 9.1, 2.8 Hz, 1H, Ar-H5`), 7.29–7.34 (m, 1H, Ar-H4`), 7.45 (s, 2H, Ar-H4/ H6), 7.49 (t, *J* = 7.6 Hz, 1H, Ar-H5), 7.54–7.58 (m, 1H, Ar-H7`), 7.95 (d, *J* = 7.4 Hz, 2H, Ar-H3/H7), 12.87 (s, 1H, NH amide).**$^{13}$C-NMR (125 MHz, DMSO)** δ (ppm): 107.9 (d, $J^2_{C-F}$ = 27.2, C7`), 112.7 (d, $J^2_{C-F}$ == 23.5 Hz, C5`), 128.5 (d, $J^3_{C-F}$ = 8.7 Hz, C4`), 129.3 (d, d, $J^3_{C-F}$ = 8.4, C7a`), 130.8 (C4/C6), 131.8 (C3/C7), 131.9 (C5), 132.9 (C2), 133.0 (C3a`), 149.5 (C2'), 157.2 (d, $J^1_{C-F}$ = 236.3 Hz, C6`), 166.3 (C1) ppm. **HRMS (ESI)** m/z: Calculated for $C_{14}H_{10}OFN_2S$ (M + H) $^+$ = 272.04196; found 273.04924. **Supplementary data charts (S# 5) in** S1 File show $^1$H-NMR, $^{13}$C-NMR, DEPT-135, and HRMS (ESI) spectrum of **compound 9**.

- ***N*-(3-benzoylphenyl) benzamide (10)**

White powder; yield (86%); Mp 120–122°C; mobile phase (EtOAc: n-hexane) (1:3); Rf = 0.37; **$^1$H-NMR (500 MHz, DMSO-d$_6$)** δ (ppm): 7.48 (br. d, 1H, Ar-H11"), 7.50–7.63 (m, 6H, Ar-H6" + H4" + H4/ H6 + H9"/ H13"), 7.69 (ps. t, 1H, Ar-H5"), 7.78 (d, *J* = 7.8 Hz, 2H, Ar-H10"/ H12"), 7.98 (d, *J* = 7.8 Hz, 2H, Ar-H3/ H7), 8.15 (ps. t, 1H, Ar-H5), 8.24 (s, 1H, Ar-H2"), 10.49 (s, 1H, NH amide) ppm. **$^{13}$C-NMR (125 MHz, DMSO-d$_6$)** δ (ppm): 121.3 (C2"), 124.2 (C6"), 124.8 (C4"), 127.7 (C4/C6), 128.4 (C3/C7),128.6 (C9"/C13"), 129.0 (C10"/C12"), 129.6 (C5), 131.8 (C5"), 132.7 (C11"), 134.6 (C1"), 137.1 (C8"), 137.4 (C2), 139.4 (C3"), 165.8 (C1), 195.6 (C7") ppm. **HRMS (ESI)** m/z: Calculated for $C_{20}H_{15}N_1Na_1O_2$ (M + Na) $^+$ = 301.11028; found 324.09950. **Supplementary data charts (S# 6) in** S1 File show $^1$H-NMR, $^{13}$C-NMR, DEPT-135, and HRMS (ESI) spectrum of **compound 10**.

- ***N*-(4-benzoylphenyl) benzamide (11)**

White powder; yield (71%); Mp 143−145°C; mobile phase (EtOAc: n-hexane) (1:3); Rf = 0.41; **$^1$H-NMR (500 MHz, DMSO-d$_6$)** δ (ppm): 7.56 (m, 4H, Ar-H4/ H6 + H10"/H12"), 7.62 (m, 1H, Ar-H11"), 7.67 (m 1H, Ar-H5), 7.73 (d, *J* = 7.5 Hz, 2H, Ar-H2"/ H6"), 7.79 (d, *J* = 8.3 Hz, 2H, Ar-H3"/H5"), 8.00 (t, *J* = 7.8 Hz, 4H, Ar-H9"/H13" + H3/H7), 10.61 (s, 1H, NH amide)

ppm. **¹³C-NMR (125 MHz, DMSO-d₆)** δ (ppm): 119.4 (C11"), 127.8 (C4/C6), 128.5 (C3/ C7), 128.5 (C10"/C12"), 129.4 (C9"/C13"), 131.0 (C2"/C6"), 131.7 (C3"/C5"), 132.3 (C2), 134.6 (C4"), 137.5 (C8"), 143.4 (C1"), 166.1 (C1), 194.6 (C7") ppm. **Supplementary data charts (S# 7) in** S1 File show ¹H-NMR, ¹³C-NMR, and DEPT-135 (DMSO-d6) spectrum of **compound 11**.

- *N*-(2-benzoylphenyl) benzamide (**12**)

Faint yellow powder; yield (78%); Mp 1128–130°C, mobile phase (CHCl₃); Rf = 0.67; **¹H-NMR (500 MHz, DMSO-d₆)** δ (ppm):7.32 (ps. t, 1H, Ar-H2"), 7.42–7.51 (m, 5H, Ar-H5 + H10"/H12" + H4/H6), 7.56 (m, 2H, Ar-H4" + H3"), 7.65 (t, *J* = 7.8 Hz, 1H, Ar-H11"), 7.63–7.70 (m, 4H, Ar-H9"/H13" + H3/H7), 7.87 (br. d, 1H, Ar-H5"), 10.77 (s, 1H, NH amide) ppm. **¹³C-NMR (125 MHz, DMSO-d₆)** δ (ppm): 123.7 (C11"), 124.2 (C5), 127.1 (C3/C7), 128.0 (C4/C6), 128.2 (C3" + C4"), 129.4 (C10"/C12"), 130.0 (C9"/C13"), 130.5 (C2"), 131.6 (C5"), 132.1 (C2), 132.3 (C8"), 134.0 (C8"), 137.0 (C1"), 137.2 (C6"), 165.1 (C1), 195.8 (C7") ppm. **HRMS (ESI)** m/z: Calculated for $C_{20}H_{15}NO_2$ (M + H) $^+$ = 301.11049; found 302.11756. **Supplementary data charts (S# 8) in** S1 File show ¹H-NMR, ¹³C-NMR, DEPT-135, and HRMS (ESI) spectrum of **compound 12**.

- *N*-(2-benzoyl-4-chlorophenyl) benzamide (**13**)

Bright yellow powder; yield (83%); Mp 120–122°C; mobile phase (EtOAc: n-hexane) (1:3); Rf = 0.30; **¹H-NMR (500 MHz, DMSO-d₆)** δ (ppm): 7.42 (ps. t, 2H, Ar-H4/ H6), 7.44–7.49 (m, 3H, Ar-H2" + H3/ H7), 7.52 (t, *J* = 7.4 Hz, 1H, Ar-H11"), 7.58 (t, *J* = 7.4 Hz, 1H, Ar-H5), 7.62 (d, *J* = 7.7 Hz, 2H, Ar-H10"/ H12"), 7.66–7.76 (m, 4H, Ar-H9"/ H13" + H3" + H5"), 10.62 (s, 1H, NH amide) ppm. **¹³C-NMR (125 MHz, DMSO-d₆)** δ (ppm): 126.9 (C2"), 127.3 (C3/C7), 128.2 (C10"/C12"), 128.3 (C4/C6), 128.6 (C9"/C13"), 129.4 (C4"), 129.5 (C5"), 131.6 (C5), 131.8 (C11"), 132.8 (C3"), 132.9 (C2), 133.8 (C6"), 135.4 (C8"), 136.6 (C1"), 165.4 (C1), 193.8 (C7") ppm. **HRMS (ESI)** m/z: Calculated for $C_{20}H_{14}Cl_1N_1Na_1O_2$ (M + Na) $^+$ = 335.07131; found 358.06053. **Supplementary data charts (S# 9) in** S1 File show ¹H-NMR, ¹³C-NMR, and DEPT-135 (DMSO-d6) spectrum of **compound 13**.

- *N*-(5-chloro-9,10-dioxo-9,10-dihydroanthracen-1-yl) benzamide (**14**)

Orange powder; yield (96%); Mp 176−178°C, mobile phase (EtOAc: n-hexane) (1:3); Rf = 0.68; **¹H-NMR (500 MHz, DMSO-d₆)** δ (ppm): 7.59–7.64 (m, 3H, Ar-H4/ H6 + H8'), 7.81 (m, 2H, Ar-H4' + H5), 7.85–7.90 (m, 3H, Ar-H3/ H7 + H9'), 8.02 (br. d, 1H, Ar-H7`), 8.12 (d, *J* = 9.4 Hz, 1H, Ar-H2`), 8.20 (m, 1H, Ar-H4'), 12.80 (s, 1H, NH amide) ppm. **¹³C-NMR (125 MHz, DMSO-d₆)** δ (ppm): 121.4 (C9`), 124.3 (C10ᵃ`), 124.4 (C4/C6), 124.6 (C2'), 126.6 (C3/C7), 128.2 (C8'), 128.9 (C6'), 128.9 (C5'a), 129.0 (C9'a), 129.8 (C6'), 130.2 (C2), 132.5 (C10'a), 132.9 (C4'a), 134.3 (C4'), 135.6 (C3'), 137.5 (C7'), 140.6 (C1'), 164.7 (C1), 185.8 (C10'), 186.2 (C5') ppm. **HRMS (ESI)** m/z: Calculated for $C_{21}H_{12}N_1Cl_1Na_1O_3$ (M + Na) $^+$ = 361.05057; found 384.03979. **Supplementary data charts (S# 10) in** S1 File **show** ¹H-NMR, ¹³C-NMR, and HRMS (ESI) spectra of **compound 14**.

- *N*-Acridin-9-yl-benzamide (**15**)

Yellow crystals; yield (79%); Mp 130–131°C, mobile phase (Column chromatography) (EtOAc: n-hexane) (20:80); Rf = 0.6; **¹H-NMR (500 MHz, DMSO-d₆)** δ (ppm): 7.50 (t, *J* = 7.6 Hz, 5H, Ar-H2`/H3` + H6`/H7` + H5), 7.62 (t, *J* = 7.4 Hz, 2H, Ar-H4/ H6), 7.94 (d, *J* = 7.7 Hz, 6H, Ar-H5`/H8` + H1`/H4` + H3/H7).**¹³C-NMR (125 MHz, DMSO-d₆)** δ (ppm): 128.6 (C2`/C3` + C6`/C7` + C5), 129.3 (C1`/C4` + C5`/C8` + C3/C7), 130.7 (C4/C6), 132.9 (C9` + C4a` + C10a` + C9a` + C8a`), 167.3 (C1). **Supplementary data charts (S# 11) in** S1 File show ¹H-NMR, ¹³C-NMR, and DEPT-135 spectrum of **compound 15**.

## General synthesis of 1H-indole-2-carboxamide scaffold

**Synthesis of 1H-indole-2-carbonyl chloride:** 1H-indole-2-carboxylic acid (0.50 g, 3.1 mmol) was weighed in a flask and dried in an oven at 80°C for 1 hr, then dissolved in an anhydrous CHCl₃ (20 mL) for 2 minutes, followed by adding oxalyl chloride (1.06 mL, 12.4 mmol, 4 eq) with stirring for 2 minutes. Then, three drops of DMF were added and stirred for 30

minutes at 0°C. The temperature was raised gradually to 80°C in an oil bath for 2 hr. Next, $CHCl_3$ and unreacted oxalyl chloride were removed with simple distillation and washed with $CHCl_3$ 4 times.

**Synthesis of carboxamide derivatives:** 1H-indole-2-carbonyl chloride (0.5 g, 2.78 mmol) in an anhydrous $CHCl_3$ (20 mL) was stirred under an ice bath with an anhydrous pyridine (0.89 mL, 11.12 mmol, 4 eq) in a single-necked round-bottom flask equipped with a reflux condenser and connected to a guard tube filled with $MgSO_4$. Then, a corresponding amine derivative (**31i-x**) was dissolved in an anhydrous $CHCl_3$ (15 mL) and added to the reaction mixture portion-wise in an equimolar (1:1) ratio relative to the nucleus, and the solution was refluxed for 24 hr. At the end of the reaction time, the solvent was removed and washed with $CHCl_3$ 4 times. The residue was redissolved in $CHCl_3$ and extracted with distilled water. The product was collected after solvent evaporation, followed by recrystallization several times with $CHCl_3$/diethyl ether/n-hexane (10:20:70 mL) [23,24].

- **N-(3-(trifluoromethyl)phenyl)-1H-indole-2-carboxamide (20)**

Brown powder; yield (88%); Mp 118−120 ºC, mobile phase (EtAc: n-hexane) (1:3); Rf = 0.58; **1H-NMR (500 MHz, DMSO-d6)** δ (ppm): 6.61 (t, 1H, Ar-H2"), 6.84 (d, *J* = 8.3 Hz, 1H, Ar-H6"), 7.05 (t, 1H, Ar-H5"), 7.13 (s, *J* = 7.6 Hz, 1H, Ar-H3), 7.23 (m, 2H, Ar-H5 + H6), 7.28 (d, *J* = 7.9 Hz, 1H, Ar-H4"), 7.48 (d, 1H, Ar-H7), 7.61 (d, 1H, Ar-H4).**13C-NMR (125 MHz, DMSO-d6)** δ (ppm): 108.1 (C2"), 111.2-111.4 (C2), 111.7−111.9 (C-$CF_3$), 113.0 (C6"), 116.2 (C7), 117.3 (C3), 120.6 (C6), 122.5 (C5), 126.3−126.4 (C4"), 126.4−126.4 (C5"), 126.8 (C3ª), 127.4−127.4 (C3"), 133.5 (C4), 137.7−137.8 (C1"), 146.4 (C7ª), 163.4 (C1`). **Supplementary data charts (S# 12)** in S1 File show 1H-NMR, 13C-NMR, and DEPT-135 spectrum of **compound 20**.

- **N-(2-(trifluoromethyl)phenyl)-1H-indole-2-carboxamide (21)**

Brown powder; yield (89%); Mp 125−127°C; mobile phase (EtAc: n-hexane) (1:3); Rf = 0.52; **1H-NMR (500 MHz, DMSO-d6)** δ (ppm): 6.49 (t, *J* = 7.6 Hz, 1H, Ar-H4"), 6.71 (d, *J* = 8.2 Hz, 1H, Ar-H6"), 6.91 (t, *J* = 7.5 Hz, 1H, Ar-H5"), 7.07−7.15 (m, 4H, Ar-H5 + H6 + H3 + H3"), 7.41 (d, *J* = 8.3 Hz, 1H, Ar-H7), 7.47 (d, *J* = 8.0 Hz, 1H, Ar-H4). **13C-NMR (125 MHz, DMSO-d6)** δ (ppm): 109.6 ($CF_3$), 117.7 (C7), 118.3 (C3), 121.6 (C6), 123.3 (C4), 125.2 (C6"), 126.2 (C5) 127.1−127.1 (C2"), 127.2 (C4"), 127.3 (C3a), 128.1−128.1 (C3"), 128.8−128.9 (C5"), 134.3 (C1"), 138.4 (C2), 146.4 (C7a), 164.8 (C1`). **Supplementary data charts (S# 13)** in S1 File show the 1H-NMR and 13C-NMR spectrum of **compound 21**.

- **N-(3-chloro-4-fluorophenyl)-1H-indole-2-carboxamide (22)**

Faint yellow powder; yield (85%); Mp 120−123 ºC, mobile phase (EtAc: n-hexane) (1:3); Rf = 0.54; **1H-NMR (500 MHz, DMSO-d6)** δ (ppm): 6.49 (d, *J* = 8.8, 3.4 Hz, 1H, Ar-H5"), 6.66 (dd, *J* = 6.4, 2.7 Hz, 1H, Ar-H6"), 6.93−7.14 (m, 3H, Ar-H2" + H3 + H5), 7.23 (t, *J* = 7.6 Hz, 1H, Ar-H6), 7.44 (d, *J* = 8.3 Hz, 1H, Ar-H7), 7.64 (d, *J* = 8.1 Hz, 1H, Ar-H4), 11.74 (s, 1H, NH indole). **13C-NMR (125 MHz, DMSO-d6)** δ (ppm): 107.3 (C7), 112.5 (C3), 113.3−113.4 (C5"), 114.1 (C6), 116.7−116.9 (C6"), 119−119.1 (C3"), 120.0 (C4), 121.9 (C5), 124.3 (C2"), 126.9 (C3a), 128.4 (C1"), 146.3−146.3 (C4"), 148.0 (C2), 149.9 (C7a), 162.8 (C1`). **Supplementary data charts (S# 14) in** S1 File show 1H-NMR, 13C-NMR, and DEPT-135 spectra of **compound 22.**

- **N-(2-chloro-4,5-dimethoxyphenyl)-1H-indole-2-carboxamide (23)**

Faint yellow powder; yield (92%); Mp 123−125°C, mobile phase (EtAc: n-hexane) (1:3); Rf = 0.57; **1H-NMR (500 MHz, DMSO-d6)** δ (ppm): 3.70 (s, *J* = 2.9 Hz, 6H, 2 $OCH_3$), 6.47 (s, 1H, Ar-H3"), 6.80 (s, 1H, Ar-H6"), 7.06 (t, *J* = 7.5 Hz, 1H, Ar-H5), 7.11 (s, 1H, Ar-H3), 7.23 (t, *J* = 7.6 Hz, 1H, Ar-H6), 7.44 (d, *J* = 8.3 Hz, 1H, Ar-H7), 7.64 (d, *J* = 8.0 Hz, 1H, Ar-H4), 11.74 (s, 1H, NH indole).**13C-NMR (125 MHz, DMSO-d6)** δ (ppm): 56.1 (O-$CH_3$), 99.3 (C6"), 105.8 (C7), 107.3 (C3"), 112.5 (C4"), 112.6 (C3), 119.9 (C6), 121.9 (C4), 124.3 (C5), 126.9 (C1"), 128.4 (C3a), 137.2 (C2), 137.8 (C7a), 140.4 (C2"), 149.0 (C5"), 162.8 (C1`). **Supplementary data charts (S# 15) in** S1 File show 1H-NMR, 13C-NMR, and DEPT-135 spectrum of **compound 23.**

- **N-(pyridin-4-yl)-1H-indole-2-carboxamide (24)**

Brown powder; yield (81%); Mp 95–98°C; mobile phase (EtOAc: n-hexane) (1:3); Rf = 0.52; **¹H-NMR (500 MHz, DMSO-d₆)** δ (ppm): 6.58 (d, J = 5.8 Hz, 2H, Ar-H3"/H5"), 6.86 (s, 1H, Ar-H3), 6.98 (t, J = 7.5 Hz, 1H, Ar-H5), 7.13 (t, J = 7.6 Hz, 1H, Ar-H6), 7.41 (d, J = 8.2 Hz, 1H, Ar-H7), 7.56 (d, J = 8.0 Hz, 1H, Ar-H4), 8.05 (d, J = 5.8 Hz, 2H, Ar-H2"/H6"), 11.38 (s, 1H, NH indole). **¹³C-NMR (125 MHz, DMSO-d₆)** δ (ppm): 104.2 (C7), 108.8 (C3"/C5"), 112.2 (C3), 119.1 (C6), 121.2 (C4), 122.6 (C5), 127.4 (C3a), 134.0 (C2), 136.4 (C7a), 146.4 (C2"/C6"), 155.9 (C4"), 165.0 (C1`). **Supplementary data charts (S# 16) in** S1 File show ¹H-NMR, ¹³C-NMR, and DEPT-135 spectrum of **compound 24.**

- **N-(pyridin-3-yl)-1H-indole-2-carboxamide (25)**

Brown powder; yield (78%); Mp 98–100°C, mobile phase (EtOAc: n-hexane) (1:3); Rf = 0.49; **¹H-NMR (500 MHz, DMSO-d₆)** δ (ppm): 6.92 (dd, 1H, Ar-H4"), 6.95–7.17 (m, 2H, Ar-H3 + H5), 7.23 (t, J = 7.6 Hz, 1H, Ar-H6), 7.45 (d, J = 8.3 Hz, 1H, Ar-H7), 7.64 (d, J = 8.0 Hz, 1H, Ar-H4), 7.74 (d, J = 4.7 Hz, 1H, Ar-H6"), 7.95 (s, J = 2.7 Hz, 1H, Ar-H2"), 11.76 (s, 1H, NH indole). **¹³C-NMR (125 MHz, DMSO-d₆)** δ (ppm): 107.1 (C7), 112.5 (C3), 119.8 (C4"), 119.9 (C6), 121.9 (C4), 123.6 (C5), 124.2 (C5"), 126.9 (C3a), 128.8 (C2), 136.2 (C2"), 136.8 (C6"), 137.2 (C7a), 144.9 (C3"), 163.0 (C1`). **Supplementary data charts (S# 17)** in S1 File show ¹H-NMR, ¹³C-NMR, and DEPT-135 spectrum of **compound 25.**

- **N-(3-amino-5-methylisoxazol-4-yl)-1H-indole-2-carboxamide (26)**

Brown powder; yield (90%); Mp 128–130°C, mobile phase (EtOAc: n-hexane) (1:3); Rf = 0.42; **¹H-NMR (500 MHz, DMSO-d₆)** δ (ppm): 5.71 (s, 3H, H6"), 5.88 (s, 1H, Ar-H4"), 7.36 (t, J = 7.5 Hz, 1H, Ar-H5), 7.40 (d, J = 2.1 Hz, 1H, Ar-H3), 7.51–7.57 (t, 1H, Ar-H6), 7.75 (d, J = 8.3 Hz, 1H, Ar-H7), 7.94 (d, J = 8.0 Hz, 1H, Ar-H4), 12.04 (s, 1H, NH indole). **¹³C-NMR (125 MHz, DMSO-d₆)** δ (ppm): 12.0 (C6"), 94.4 (C7), 107.3 (C4"), 112.5 (C3), 120.0 (C6), 122.0 (C4), 124.3 (C5), 126.9 (C3a), 128.6 (C2), 137.3 (C7a), 162.9 (C3"), 164.1 (C5"), 167.5 (C1`). **Supplementary data charts (S# 18) in** S1 File show ¹H-NMR, ¹³C-NMR, and DEPT-135 spectra of **compound 26.**

- **N-[(4-hydroxyphenyl)methyl]-1H-indole-2-carboxamide (27)**

Yellow powder; yield (89%); Mp 120–122°C; mobile phase (EtOAc: n-hexane) (1:3); Rf = 0.53; **¹H-NMR (500 MHz, DMSO-d₆)** δ (ppm): 3.81 (s, 2H, H3`), 6.66 (s, 1H, Ar-H3), 6.76 (d, J = 7.9 Hz, 2H, Ar-H3"/H5"), 6.93 (t, J = 7.4 Hz, 1H, Ar-H5), 7.05 (t, J = 7.6 Hz, 1H, Ar-H6), 7.22 (d, J = 8.0 Hz, 2H, Ar-H2"/H6"), 7.37 (d, J = 8.2 Hz, 1H, Ar-H7), 7.50 (d, J = 7.9 Hz, 1H, Ar-H4), 11.01 (s, 1H, NH indole). **¹³C-NMR (125 MHz, DMSO-d₆)** δ (ppm): 43.0 (C3`), 102.3 (C7), 112.0 (C3), 115.2 (C3"/C5"), 118.6 (C6), 120.8 (C4), 121.6 (C5), 127.8 (C2"/C6"), 127.9 (C3a), 129.6 (C1"), 135.8 (C2), 137.5 (C7a), 157.2 (C4"), 165.8 (C1`). **Supplementary data charts (S# 19)** in S1 File show ¹H-NMR, ¹³C-NMR, and DEPT-135 spectrum of **compound 27.**

- **N-(5-amino-1H-imidazol-4-yl)-1H-indole-2-carboxamide (28)**

Yellow powder; yield (93%); Mp 168–170°C; mobile phase (EtOAc: n-hexane) (1:3); Rf = 0.40; **¹H-NMR (500 MHz, DMSO-d₆)** δ (ppm): 6.73 (s, 2H, NH2), 7.05 (t, J = 7.5 Hz, 1H, Ar-H5), 7.09 (S, J = 2.2 Hz, 1H, Ar-H2"), 7.15 (s, 1H, Ar-H3), 7.23 (t, J = 7.6 Hz, 1H, Ar-H6), 7.44 (d, J = 8.3 Hz, 1H, Ar-H7), 7.64 (d, J = 8.0 Hz, 1H, Ar-H4), 11.74 (s, 1H, NH indole). **¹³C-NMR (125 MHz, DMSO-d₆)** δ (ppm): 107.3 (C7), 108.9 (C3), 112.5 (C6), 120.0 (C4), 122.0 (C5), 124.3 (C3a), 126.9 (C5"), 128.6 (C4"), 129.6 (C2), 137.3 (C7a), 146.2 (C2"), 163.0 (C3`), 165.6 (C1`). **Supplementary data charts (S# 20)** in S1 File show the ¹H-NMR and ¹³C-NMR spectrum of **compound 28.**

- **N-(4-methylphenyl)-1H-indole-2-carboxamide (29)**

Brown powder; yield (88%); Mp 155–159°C, mobile phase (EtOAc: n-hexane) (1:3); Rf = 0.69; **¹H-NMR (500 MHz, DMSO-d₆)** δ (ppm): 2.24 (s, 3H, CH₃), 6.92–6.98 (m, 1H, Ar-H3), 7.08 (d, J = 8.0 Hz, 2H, Ar-H2"/H6"), 7.27 (t, J = 7.7 Hz, 2H, Ar-H5/H6), 7.33 (d, J = 8.0 Hz, 2H, Ar-H3"/H5"), 7.44 (d, J = 7.9 Hz, 2H, Ar-H4/H7), 8.53 (s, 1H, NH amide), 8.59 (s,

1H, NH indole).<sup>13</sup>**C-NMR (125 MHz, DMSO-d$_6$)** δ (ppm): 20.3 (CH$_3$), 118.1 (C5/C6), 118.1 (C4/C7), 118.2 (C2''/C6''), 121.7 (C3), 128.7 (C3''/C5''), 129.1 (C1''+C4''), 130.6 (C3a), 137.1 (C2), 139.8 (C7a), 152.5 (C1`). **HRMS (ESI)** m/z: Calculated for C$_{16}$H$_{14}$N$_2$O (M+K)$^+$ = 250.11081; found 289.07377. **Supplementary data charts (S# 21)** S1 File show the <sup>1</sup>H-NMR, <sup>13</sup>C-NMR, and HRMS (ESI) spectrum of **compound 29.**

### In-Vitro method

**MMP-9 inhibitors screening assay.** The MMP-9 inhibitory potential of the studied compounds was evaluated using the colorimetric inhibitor screening assay kit protocol (Abcam, Catalog No. ab139448). Initially, we carried out a screening step for all the studied compounds to determine the percentage inhibition at a single concentration of 50 μM. Briefly, the tested compound and the reference inhibitor (NNGH) were added to the assay buffer and the MMP-9 enzyme, and then incubated for 60 min at 37 ℃. After that, the chromogenic substrate (Ac-PLG [2-mercapto-4-methyl-pentanoyl]-LG-OC2H5) was added, and the absorbance at 412 nm was immediately measured every minute for 20 min. The inhibition percentage of MMP-9 for each tested compound was calculated, as shown in Equation 1.

$$\% \text{ inhibition } = 100 - (\text{velocity of compound}/ \text{velocity of control}) * 100$$

Equation 1**.** The percent inhibition equation is used to determine MMP-9 enzymatic activity.

The compounds with the highest inhibition percent (>60%) were evaluated at different concentrations (100, 50, 25, and 12.5 μM) to determine their IC$_{50}$ values following the exact instructions. The IC$_{50}$ values were calculated using GraphPad Prism.

## Results and discussion

### Chemistry

**Synthesis of benzamide compounds.** Based on the pharmacophoric features and docking scores, a series of benzamide compounds was synthesized via a nucleophilic acyl substitution mechanism. The compounds **5–15** were synthesized by reacting benzoyl chloride (**3**) (0.5 g, 3.5 mmol) as a nucleus with corresponding amines (**4i-xi**) (1:1 eq) using an anhydrous CHCl$_3$ and pyridine under reflux, as stated in Fig 1 and Fig 2. The chemical structures of targeted compounds were characterized using <sup>1</sup>H-NMR, <sup>13</sup>C-NMR, DEPT-135, and HRMS (ESI). In <sup>13</sup>C-NMR spectra, the carbonyl amide appeared in the range of 160–175 ppm. In <sup>1</sup>H-NMR, the aromatic proton signals appeared from 6.3 to 8.2 ppm. All relevant data, along with the synthesized structures, are provided in the experimental section. Due to the solubility difficulties, representative MS was performed in the present study.

**Synthesis of 1H-indole-2-carboxamide compounds.** A series of 1H-indole-2-carboxamide compounds was synthesized by a two-step reaction. The first step involves the activation of 1H-indole-2-carboxylic acid (**16**) by oxalyl chloride (**17**) (1:4 eq) in the presence of a few drops of DMF as a catalyst and dry CHCl$_3$ as a solvent, which led to the formation of 1H-indole-2-carbonyl chloride (**18**) under a reflux system as shown in Fig 3. Compound **18** was collected after the disappearance of the 1H-indole-2-carboxylic acid (**16**) spot on TLC. The target compounds **20–29** were synthesized by

**Fig 2. Synthesis of benzamide compounds.** Reagents and conditions: (a) CHCl$_3$, pyridine, reflux at 80°C for 24 hr.

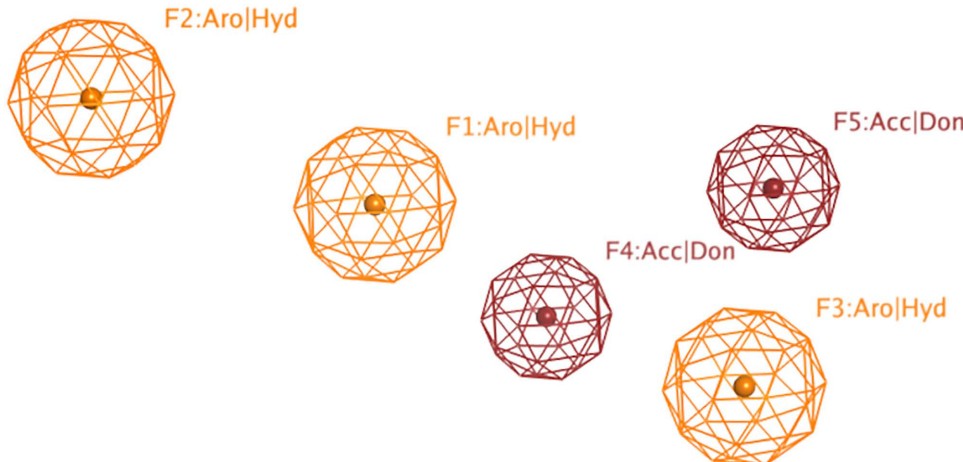

**Fig 3. Synthesis of 1H-indole-2-carboxamide derivatives.** Reagents and conditions: (a) (1) CHCl₃, DMF, 0 °C, 30 min, (2) Reflux at 80 °C for 2 hr (b) CHCl₃, pyridine, reflux at 80 °C for 24 hr.

reacting **18** with the corresponding amines (**19i-x**) (1:1 eq) using anhydrous CHCl₃ and pyridine under reflux, as stated in Fig 1 and Fig 3. The chemical structures of targeted compounds were characterized using ¹H-NMR, ¹³C-NMR, DEPT-135, and HRMS (ESI). In ¹³C-NMR spectra, the carbonyl amide appeared in a range of 152–169 ppm. In ¹H-NMR, the aromatic proton signals appeared from 6.2 to 8.4 ppm. All relevant data, along with the synthesized structures, are provided in the experimental section. Due to the solubility difficulties, representative MS was performed in the present study.

## Pharmacophore modeling

We adopted the coordinates of the 7MR template in MMP-9 (PDB ID: 2OW1) to draw the 3D structures of the verified selective MMP-9 inhibitors (48 inhibitors from different databases, including Google Scholar, PubMed, PubChem, Cortellis database, and Protein Data Bank (PDB) (S1 Table) [25,26]. The refined flexible alignment of the ligands coordinates against 7MR (**Compound xliii**) produced a pharmacophore model with five features (Fig 4). F1, F2, and F3 stand for aromatic or hydrophobic moiety (Aro/HY), and F4 and F5 stand for H-Bond donor (HBD) or acceptor (HBA). The produced

**Fig 4. Pharmacophoric features of MMP-9 selective inhibitors.** Aro stands for an aromatic ring, Hyd: hydrophobic, Acc: acceptor, Don: donor functional groups. Picture made by MOE [15].

model indicates that MMP-9 inhibitors should harbor three aromatic rings or hydrophobic moieties and two H-bond acceptors or donors. Fortunately, 41 out of 43 ligands were found to match the five pharmacophoric features.

According to the literature, several pharmacophoric models have been developed, revealing the significance of HBD, HBA, and Aro/HY features in the MMP-9 model. Kalva S et al. [27] developed a pharmacophoric model composed of HBA, HBD, and Aro ring, which was used to retrieve new MMP-9 inhibitors. Additionally, Rathee et al. developed a pharmacophore model and 3D-QSAR model to predict the MMP-9 inhibitory activity of hydroxamate scaffolds. The five-point (AAARR) pharmacophoric model included three HBA features and two aromatic rings [28]. Additionally, three ligand-based pharmacophore models were developed based on known inhibitors of MMP-9. The best model consists of two HBAs, one HBD, one HY, and one aromatic ring feature [29]. In 2021, Sanapalli et al. developed and validated a robust five-point 3D-QSAR model that includes two HBD features, one HY feature, and two Aro ring features. Using virtual screening and molecular docking methods, two compounds were identified as selective inhibitors of MMP-9 [30]. These results highlight the importance of HBD, HBA, and Aro/HY functional groups in MMP-9 inhibitors for their ability to fit into the MMP-9 binding pocket.

## Pharmacophore screening

We screened the generated pharmacophore model against the NCI database, which contains 265,242 compounds [31]. The NCI database was filtered using the Lipinski rule of five to retrieve drug-like molecules (except for M.WT up to 600) [32]. The dataset was filtered to include only compounds with molecular weights within 200–600 g/mol, log P values ≤ 5, HBD ≤ 5, and HBA ≤ 10. Then, two methods of hit searching were applied (Fig 5). The first one searched for hits that partially matched the pharmacophoric features (at least four features), and the second one searched for hits that fully matched the pharmacophoric features. Fortunately, 125,740 and 4,350 hits were obtained from the partial and full match methods, respectively. Further filtration of these hits was carried out using a virtual screening method by MAESTRO, based on Glide docking scores in kcal/mol. The more negative the docking score, the better the binder.

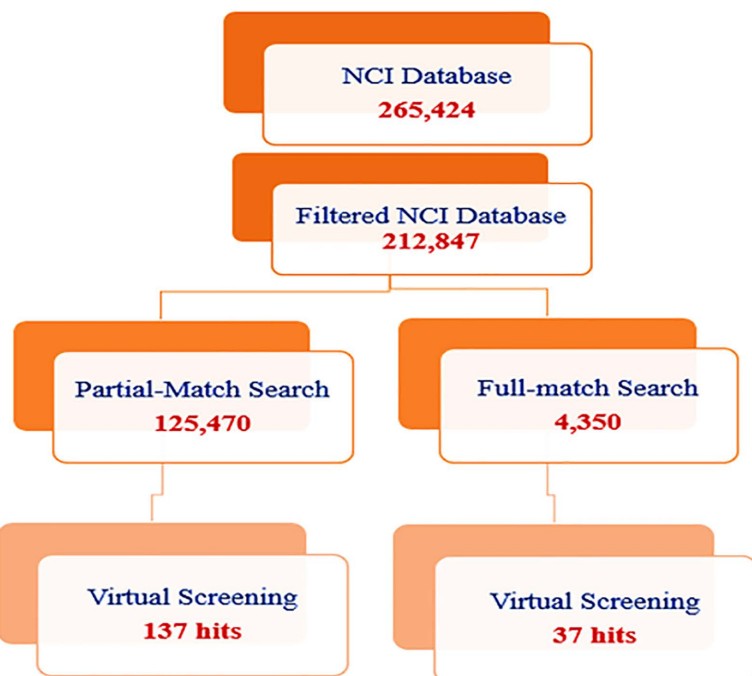

**Fig 5. The pharmacophore search workflow.**

## Full match hits-based virtual screening

Based on the virtual screening, 37 hits were fetched from the screening of fully matched hits (S2 Table). Fig 6 shows the fitting of five functional groups of NSC32903 and NSC116564 with the five MMP-9 pharmacophoric features.

## Partial match hits-based virtual screening

Based on the virtual screening of partially matched hits, 137 hits were produced (S3 Table). Two hits were purchased from the partially matched list (**NSC3276** and **NSC123019**) to be examined using the colorimetric MMP-9 inhibitor activity screening assay (Fig 7).

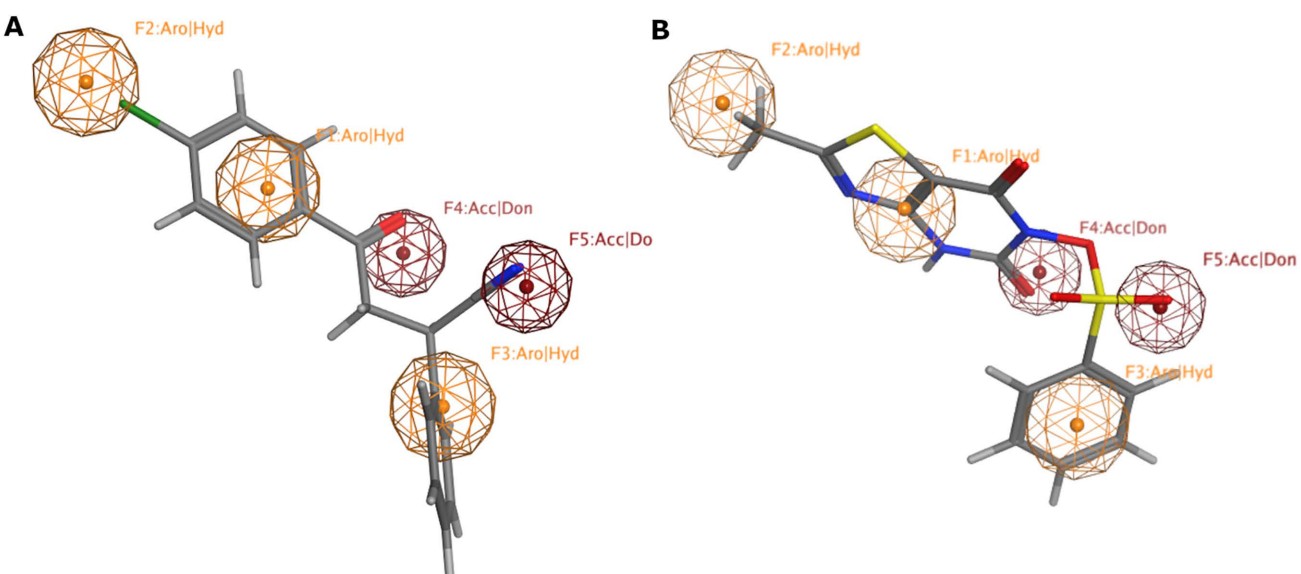

**Fig 6. MMP-9 pharmacophore model with full match hits: (A) NSC32903, (B) NSC116564.** Aro stands for the aromatic ring, Hyd: hydrophobic, Acc: acceptor, Don: donor groups. Picture made by MOE [15].

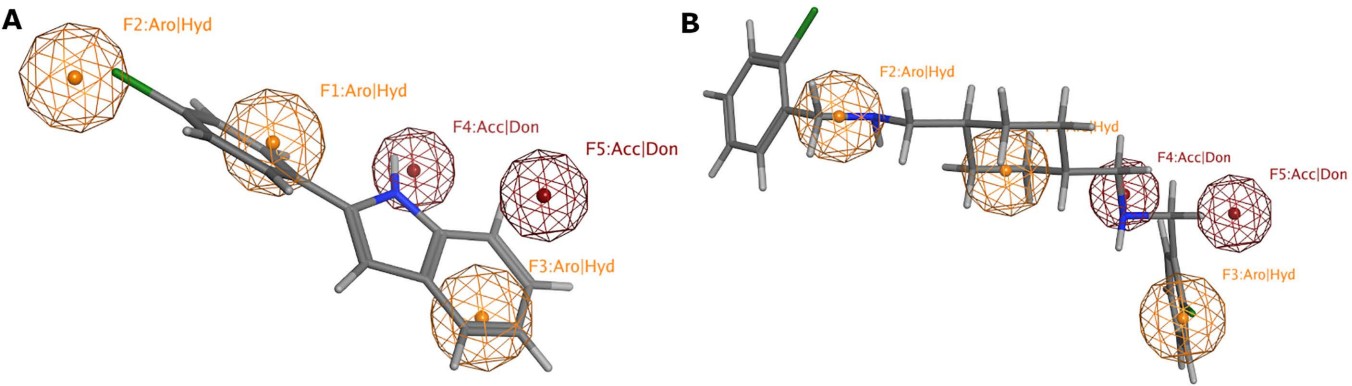

**Fig 7. MMP-9 pharmacophore model with purchased-partial match hits: (A) NSC3276, (B) NSC123019.** Aro stands for the aromatic ring, Hyd: hydrophobic, Acc: acceptor, Don: donor group. Picture made by MOE [15].

## Molecular docking

In order to explain the potential cytotoxicity of the verified compounds, molecular docking studies were performed against the coordinates of WT MMP-9 (PDB ID: 1GKC) [18] and MUT MMP-9 (PDB ID: 2OW1) [16] to assess the structural basis of MMP-9s/ligand interaction in the MMP-9 binding site through Glide and induced-fit docking (IFD) studies [33,34].

The Glide (grid-based ligand docking energetics) is a comprehensive algorithm for the ligand/ receptor docking strategy [34]. It offers a high-throughput virtual screening process of millions of compounds with high speed and accuracy. Moreover, the Glide score has been implemented with an "extra precision" (XP) scoring function [34]. This protocol probes water desolvation energy and receptor-ligand structural backbones, consequently improving the binding affinity of complex formation [33]. The IFD approach explores the protein structural changes in this way: ligands are docked to a protein`s binding site through Glide docking [35] and the prime algorithm is used to minimize the best ligand conformation along with the MMP-9 binding site [36]. Then, a redocking protocol is conducted against the minimized form of MMP-9. Subsequently, protein flexibility is considered during the docking procedure.

To explore the structure-basis of binding of all the reported MMP-9 active inhibitors in literature, we carried out Glide and IFD studies against WT (1GKC) and MUT (2OW1) MMP-9 coordinates. Most of the inhibitors fit the 1GKC and 2OW1 binding sites, and most of them form at least one H bond. The Glide and IF docking score (kcal/mol) and binding interactions of all the reported inhibitors are shown in S4 and S5 Tables, respectively.

The binding site of WT MMP-9 (1GKC) and MUT MMP-9 (2OW1) with their co-crystalized ligands NFH and 7MR is shown in Fig 8A and B, respectively. The only difference between the WT and MUT MMP-9 is revealed in one amino acid residue (sequence # 402); in 1GKC, glutamic acid is present (E 402) while in 2OW1, glutamine is shown (Q 402).

The binding pocket of wild MMP-9 (1GKC) encloses GLY186, LEU187, TYR393, VAL398, GLU402, HIS405, HIS411, TYR420, MET422, ARG424, and PHE425 (Fig 6 A). The docking studies against 1GKC showed that compounds form π-cation, π-π stacking, hydrophobic, and H-bond with the backbones of key binding residues. In addition, one coordination bond with Zn$^{+2}$ takes place. Particularly, π-π stacking engages the compounds with PHE110, HIS401, HIS411, and TYR423. H-bond binds the compounds with TYR179, GLY186, LEU187, LEU188, ALA189, TYR420, PRO421, MET422, TYR423, and ARG424.

Interestingly, other computational and experimental studies pinpointed the importance of these amino acids as key binding residues for MMP-9/ ligand interactions [38,39]. Reported docking studies against 1GKC showed that the compounds form H-bond and π-π stacking with the backbone of key binding residues, particularly with PRO421 and MET422 [38]. Interestingly, another study emphasizes the significance of TYR423 and Met422 residues in 1GKC/ ligand complex formation through specific H-bonds. A π-π interaction with TYR420 reinforces the binding to the MMP-9 catalytic domain

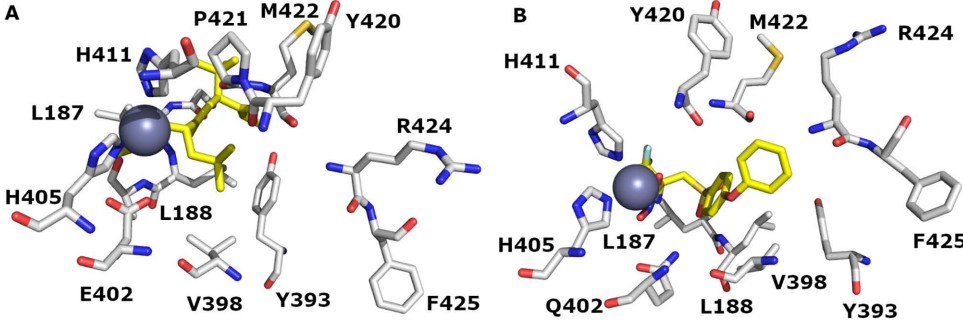

**Fig 8. The binding domains of (A) WT MMP-9 (PDB ID: 1GKC) and (B) MUT MMP-9 (PDB ID: 2OW1) accommodate the verified series.** Carbon (C) atoms are depicted in grey color, N blue color, and O red color. H bonds and valence were removed for clarification. Zinc metal is presented as a sphere (metallic color). Picture visualized by PYMOL [37].

[39]. Furthermore, analysis of the experimental crystal structure of MMP-9 highlights the significance of Gly186, LEU188, ALA191, Glu402, PRO421, and TYR423 residues in the 1GKC binding pocket [18]. All these docking studies emphasize the significance of these amino acids in the 1GKC binding domain and their importance in MMP-P/ligand complex formation, which is comparable to our Glide and IF docking studies results.

Additionally, Glide and IF docking studies were conducted for the reported MMP-9 inhibitors against the MUT MMP-9 (2OW1). We found that 2OW1 encloses GLY186, LEU187, TYR393, VAL398, GLN402, HIS405, HIS411, TYR420, MET422, ARG424, and PHE425 (Fig 8 B). All of the docked compounds occupied the 2OW1 binding domain. The docking studies against 2OW1 showed that compounds form π-π stacking, hydrophobic, and H-bonds with the backbones of key binding residues. In addition, one coordination bond with Zn$^{+2}$ takes place. Particularly, π-π stacking engages the compounds with PHE110, TYR179, HIS401, HIS411, and TYR423. H-bond binds the compounds with LEU188, ALA189, HIS190, LEU397, GLN402, ASP 410, HIS411, LEU 418, TYR420, PRO421, MET 422, and ARG 424. Additional halogen bonds were found to bind the compounds through HIS411, ALA 417, and ARG 424.

The importance of these amino acids in the 2OW1 binding domain was previously reported [16,40–43]. Among the broad classes of MMP-9 inhibitors [40,41]: carboxylic acid-based inhibitors, sulfonamide hydroxamate-based inhibitors, and thioester-based inhibitors, a set of 54 inhibitors was chosen for docking studies to get a structural insight into the MMP-9 binding pocket (PDB: 2OW1) [42]. They found that the S1 Table`binding pocket is composed of ASP185-LEU188 and PRO421-TYR423, which act as HBD and HBA. The lining of this pocket comprises LEU188, LEU397, Val 398, HIS401, Leu 418, and Met 422-TYR423. LEU397 and Val 398 were found to be specific to MMP-9 [16]. Metal coordination bonds were facilitated between the substrate functional groups and the zinc metal resulting in anchoring the ligand binding group with S1 Table` cavity residues. The best docking conformations comprise either a H-bond with LEU188-ALA189 in the upper domain or GLN402-HIS411 in the lower domain. Indeed, they found that the bulkier functional group prefers binding in the LEU188. While the smaller ligand moieties showed a preference for GLN402-HIS411 in the lower domain. Regarding the S1 Table` lining residues (PRO421-TYR423), it acts as an HBA with the ligand functional groups. These results confirm that S1` cavity is the most important ligand recognition point in the binding site [42].

Another docking study of hydroxamic acid inhibitors was conducted to investigate the inhibition of the MMP-9 binding pocket (PDB:2OW1) [16,43]. The compounds showed two H-bonds with the backbones of key binding residues. Particularly, H-bond engages inhibitors with Leu 187, ALA189, GLN402, and PRO421. Regarding the ligand hydrophobic group, it was found to be extended out of the cleft and form a van der Waals contact with the side chains of PRO421, HIS411, and LEU187 [43]. These docking studies revealed the importance of these residues in the 2OW1 binding domain, which is similar to the residues highlighted by our Glide and IFD studies.

## Molecular docking of synthesized compounds and hits

Parallel docking studies were performed for two classes of synthesized compounds, including benzamide and 1H-indole-2-carboxamide scaffolds. Glide and IFD docking approaches were performed against both 1GKC and 2OW1. All the compounds fit the 1GKC and 2OW1 binding domains. The Glide and IFD scores (kcal/mol) and binding interactions of all the synthesized compounds are shown in Table 1.

## Benzamide scaffolds

The synthesized benzamide compounds were docked to the 1GKC and 2OW1 MMP-9 binding pockets. All compounds fit 1GKC and 2OW1 binding domains as shown in Fig 9A. The IF docked pose of compound **8** superposed the co-crystallized ligand (NFH) as shown in Fig 9 B. Compound **8** shows the highest IFD score (−11.96 kcal/mol) and forms one hydrogen bond with ARG 424. While the Glide docking of compound **8** gives a −9.79 kcal/mol docking score and forms two hydrogen bonds with LEU188 and PRO421.

**Table 1. The Glide and induced-fit docking scores (Kcal/mol) and interaction forces of synthesized compounds against 1GKC and 2OW1.**

| | | 1GKC | | 2OW1 | | 1GKC | | 2OW1 | |
|---|---|---|---|---|---|---|---|---|---|
| No. | Code | Glide docking score (kcal/mol) | Interaction Forces with 1GKC | Glide docking score (kcal/mol) | Interaction Forces with 2OW1 | IFD score (kcal/mol) | Interaction Forces with 1GKC | IFD score (kcal/mol) | Interaction Forces with 2OW1 |
| 1. | Compound 8 | −9.79 | -Two H-bonds with LEU188 and PRO421 | −9.22 | -one metal coordination with $Zn^{+2}$ ion - one π-π stacking with PHE110 | −11.96 | -one metal coordination with $Zn^{+2}$ ion -one π-π stacking with HIS401 -one H-bond with ARG 424 | −10.54 | -one metal coordination with $Zn^{+2}$ ion - One π-π stacking with PHE110 -Two H-bond with GLN402 |
| 2. | Compound 27 | −9.09 | -three H-bonds with TYR420 and GLY 186 | −6.01 | -Three π-π stacking with HIS401, TYR423 -One H-bond with PRO421 | −11.10 | -two π-π stacking with HIS401 -Four H-bonds with PRO421, LEU188, and TYR 393. | −8.92 | -Three π-π stacking with HIS401, TYR423 -One H-bond with PRO421 |
| 3. | Compound 20 | −9.25 | -one π-cation bond with $Zn^{+2}$ ion -two π-π stacking bonds with HIS401and TYR423 | −6.10 | -Two π-π stacking with HIS401 and TYR423 -one H-bond with PRO421 | −10.73 | Two π-π stacking with HIS401 and TYR423 -one H-bond with TYR420 | −8.43 | -Two π-π stacking with HIS401 and TYR423 -one H-bond with PRO421 |
| 4. | Compound 9 | −8.25 | -two π-π stacking with HIS401 -one H-bond with TYR420 | −8.44 | -one π-π stacking with HIS401 -one H-bond with PRO421 | −10.25 | -two π-π stacking with HIS401 -one H-bond with TYR420 | −9.62 | -one π-π stacking with HIS401 -Two H-bond with PRO421 |
| 5. | Compound 29 | −9.70 | -one π-cation bond with $Zn^{+2}$ ion -two π-π stacking bonds with HIS401 -one H-bond with TYR420 | −6.12 | -one π-cation bond with $Zn^{+2}$ ion -Two π-π stacking with HIS401 and TYR423 -one H-bond with PRO421 | −10.19 | -one π-cation bond with $Zn^{+2}$ ion -four π-π stacking with HIS401, HIS 405, and HIS411 -Two H-bonds with GLU 402 | −9.54 | -one π-cation bond with $Zn^{+2}$ ion -Two π-π stacking with HIS401 and TYR423 -one H-bond with PRO421 |
| 6. | Compound 22 | −6.32 | -one π-cation bond with $Zn^{+2}$ ion -two π-π stacking bonds with HIS401 and TYR423 -one H-bond with PRO421 | −5.45 | -one π-π stacking with HIS401 -One H-bond with PRO421 | −9.66 | -three π-π stacking with HIS401 and TYR423 -One metal coordination with $Zn^{+2}$ 444 -One H-bond with PRO421 | −7.94 | -one π-π stacking with HIS401 -One H-bond with PRO421 |
| 7. | Compound 23 | −8.30 | -two π-π stacking bonds with HIS401 and TYR423 -Two H-bonds with TYR420 and TYR423 | −6.43 | -Three π-π stacking with HIS401, HIS411, TYR423 -One metal coordination with $Zn^{+2}$ 444 -Two H-bonds with PRO421, TYR420 | −9.54 | -one π-π stacking with HIS401 -Two H-bonds TYR420 and TYR423 | −8.37 | -Two π-π stacking with HIS401 and HIS411 -One metal coordination with $Zn^{+2}$ 444 -Two H-bonds with PRO421, TYR420 |
| 8. | Compound 25 | −8.80 | -one H-bond with TYR420 | −6.34 | -two π-π stacking with HIS401 and TYR423 -one H-bond with PRO421 | −9.21 | -one H-bond with TYR420 | −8.9 | -two π-π stacking with HIS401 and TYR423 -one H-bond with PRO421 |

*(Continued)*

**Table 1.** (Continued)

| No. | Code | 1GKC Glide docking score (kcal/mol) | Interaction Forces with 1GKC | 2OW1 Glide docking score (kcal/mol) | Interaction Forces with 2OW1 | 1GKC IFD score (kcal/mol) | Interaction Forces with 1GKC | 2OW1 IFD score (kcal/mol) | Interaction Forces with 2OW1 |
|---|---|---|---|---|---|---|---|---|---|
| 9. | Compound 28 | −9.98 | -one metal coordination with $Zn^{+2}$ ion<br>-two π-π stacking with HIS401 and TYR423<br>-four H bonds with PRO421, ALA189, and LEU188 | −7.34 | -two π-π stacking with HIS401 and TYR423<br>-four H-bonds with ALA189, LEU188, and PRO421 | −9.20 | -one metal coordination with $Zn^{+2}$ ion<br>-two π-π stacking with HIS401 and TYR423 | −8.73 | -two π-π stacking with HIS401 and TYR423<br>-four H-bonds with ALA189, LEU188, and PRO421 |
| 10. | Compound 24 | −9.58 | -one π-π stacking with HIS401<br>-one H-bond with MET 422 | −6.11 | -Three π-π stacking with HIS401, TYR423, and HIS411<br>-one H-bond with PRO421 | −9.20 | -two H-bonds with TYR420 and LEU188 | −7.52 | -Three π-π stacking with HIS401, TYR423, and HIS411<br>-Two H-bonds with PRO421 and GLN402 |
| 11. | Compound 26 | −8.72 | -two metal coordination with $Zn^{+2}$<br>-two π-π stacking with HIS401 and TYR423<br>-One H-bond with TYR420 | −7.41 | -one metal coordination with $Zn^{+2}$ ion<br>-Three π-π stacking with HIS401, HIS411, and TYR423<br>-two H-bonds with PRO421 and TYR420 | −8.96 | -one metal coordination with $Zn^{+2}$ ion<br>-one π-π stacking with HIS401<br>-two H-bonds with PRO421 | −8.77 | -one metal coordination with $Zn^{+2}$ ion<br>-Three π-π stacking with HIS401, HIS411, and TYR423<br>-two H-bonds with PRO421 and TYR420 |
| 12. | Compound 10 | −8.78 | -one metal coordination with $Zn^{+2}$ ion<br>-one π-π stacking with HIS401 | −7.00 | -one metal coordination bond with $Zn^{+2}$ ion<br>-two π-π stacking with HIS401 and TYR423<br>-two H-bonds with GLN402 and ALA191 | −8.94 | -one metal coordination with $Zn^{+2}$ ion<br>-one π-π stacking with HIS401<br>-one H-bond with MET 422 | −8.91 | -one metal coordination bond with $Zn^{+2}$ ion<br>-two π-π stacking with HIS401 and TYR423<br>-Three H-bonds with GLN402 and ALA191 |
| 13. | Compound 11 | −7.40 | -one metal coordination with $Zn^{+2}$ ion<br>-one π-π stacking with HIS401 | −7.87 | -One π-cation bond with $Zn^{+2}$ ion<br>-two H-bonds with LEU188 and ALA189 | −8.61 | -two H-bonds with LEU188 and PRO421 | −8.98 | -One π-cation bond with $Zn^{+2}$ ion<br>-two H-bonds with LEU188 and ALA189 |
| 14. | Compound 21 | −8.14 | -two π-π stacking bonds with HIS401 and TYR423 | −5.48 | -Two π-π stacking with HIS401<br>-One metal coordination with $Zn^{+2}$ 444<br>-One H-bond with PRO421 | −8.35 | -one metal coordination with $Zn^{+2}$ ion<br>-Three π-π stacking with HIS401 and TYR423<br>-one H-bond with TYR420 | −7.82 | -One π-π stacking with HIS401<br>-One metal coordination with $Zn^{+2}$ 444<br>-One H-bond with PRO421 |
| 15. | Compound 14 | −7.19 | -one metal coordination with $Zn^{+2}$ ion<br>-one H-bond with TYR423<br>-one halogen bond with TYR423 | −4.58 | -One metal coordination with $Zn^{+2}$ ion<br>-Two H-bonds with ALA191 and GLN402 | −8.25 | -one metal coordination with $Zn^{+2}$ ion<br>-one H-bond with TYR423<br>-one halogen bond with TYR423 | −6.63 | -One metal coordination with $Zn^{+2}$ ion<br>-Three H-bonds with ALA191 and GLN402 |

*(Continued)*

**Table 1.** (Continued)

| | | 1GKC | | 2OW1 | | 1GKC | | 2OW1 | |
|---|---|---|---|---|---|---|---|---|---|
| No. | Code | Glide docking score (kcal/mol) | Interaction Forces with 1GKC | Glide docking score (kcal/mol) | Interaction Forces with 2OW1 | IFD score (kcal/mol) | Interaction Forces with 1GKC | IFD score (kcal/mol) | Interaction Forces with 2OW1 |
| 16. | Compound 5 | −6.35 | -three π-π stacking with HIS401 and HIS411 -one π-cation bond with $Zn^{+2}$ ion -one H-bond with Glu 402 | −7.11 | -one metal coordination bond with $Zn^{+2}$ ion -two π-π stacking with HIS401 and TYR423 -two H-bonds with GLN402 and TYR420 | −8.24 | -one π-cation bond with $Zn^{+2}$ ion -Five π-π stacking with HIS401, HIS411, and TYR423 -Two H-bonds with TYR420 and GLU 402 | −8.44 | -one metal coordination bond with $Zn^{+2}$ ion -two π-π stacking with HIS401 and TYR423 -Three H-bonds with GLN402 and TYR420 |
| 17. | Compound 13 | −7.30 | -three π-π stacking with HIS401 and HIS411 -three H-bonds with ALA189, LEU188, and PRO421 -one halogen bond with ALA191 | −4.50 | -one metal coordination bond with $Zn^{+2}$ ion -two H-bonds with GLN402 and ALA191 | −7.52 | -two π-π stacking with HIS401 -three H-bonds with PRO421, LEU188, and ALA189 | −7.54 | -one metal coordination bond with $Zn^{+2}$ ion -two H-bonds with GLN402 and ALA191 |
| 18. | Compound 7 | −7.57\ | -one metal coordination with $Zn^{+2}$ ion -one π-π stacking with HIS401 -two H-bonds with TYR423 and GLU 402 | −5.11 | -one metal coordination bond with $Zn^{+2}$ ion -two π-π stacking with HIS411 -one H-bond with GLN402 | −7.22 | -one metal coordination bond with $Zn^{+2}$ ion -two H-bonds with ALA189 and TYR423 | −6.62 | -one metal coordination bond with $Zn^{+2}$ ion -two π-π stacking with HIS411 -Two H-bonds with GLN402 |
| 19. | Compound 15 | −6.16 | -one π-cation bond with $Zn^{+2}$ ion -two π-π stacking with HIS411 and HIS401 -Two H-bonds with LEU188 and PRO421 | −5.30 | -one metal coordination with $Zn^{+2}$ ion -one π-π stacking with HIS411 | −7.20 | -one metal coordination bond with $Zn^{+2}$ ion -two π-π stacking with HIS 405 and HIS411 | −7.83 | -one metal coordination with $Zn^{+2}$ ion -one π-π stacking with HIS411 -One H-bond with LEU188 |
| 20. | Compound 12 | −7.90 | -two π-π stacking with HIS401 and HIS411 -Three H-bonds with ALA189, LEU188, and PRO421. | −4.48 | -one metal coordination bond with $Zn^{+2}$ ion -two H-bonds with GLN402 and ALA191 | −7.12 | - two π-π stacking with HIS401 and HIS411 -two H- bonds with GLU402 and ALA191 | −6.55 | -One metal coordination bond with $Zn^{+2}$ ion -Two H-bonds with GLN402 and ALA191 |
| 21. | Compound 6 | −6.42 | -one π-π stacking with TYR423 | −5.13 | -one metal coordination bond with $Zn^{+2}$ ion -two π-π stacking with HIS411 -One H-bond with GLN402 | −6.79 | -one metal coordination with $Zn^{+2}$ ion -two π-π stacking with HIS401. -one H-bond with PRO421 | −7.23 | -one metal coordination bond with $Zn^{+2}$ ion -two π-π stacking with HIS411 -Two H-bond with GLN402 |

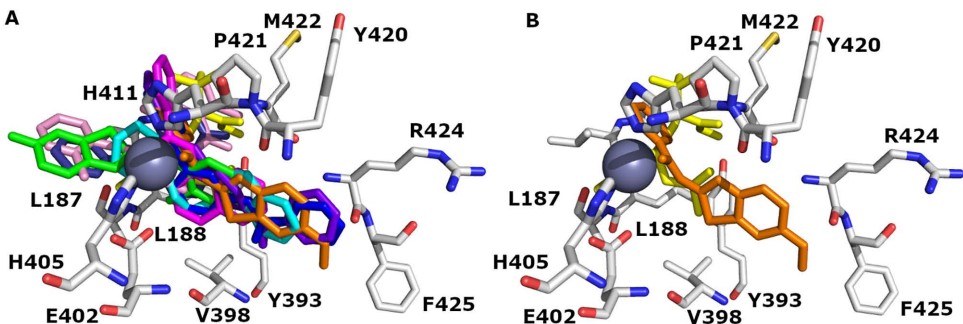

**Fig 9. The binding domains of wild MMP-9 (PDB ID: 1GKC).** (A) Orientation of IF docked poses of benzamide compounds and (**B**) overlaying of the IF docked pose of **compound 8** with the co-crystallized ligand (NFH) (yellow color). Carbon (C) atoms are depicted in grey colors, N in blue color, and O in red color. **Compound 5** depicted in cyan color, **compound 6** magenta color, **compound 8** in orange color, **compound 9** in green color, **compound 10** in blue color, **compound 11** in purple-blue color, **compound 14** in pink color, **compound 7** in deep blue color, **compound 13** in deep cyan color, **compound 12** in sand color, and **compound 15** in red color. H bonds and valence were removed for clarification. Zinc metal is presented as a sphere (metallic color). Picture visualized by PYMOL [37].

## 1H-Indole-2-carboxamide scaffolds

The synthesized 1H-Indole-2-carboxamide compounds were docked against the 1GKC and 2OW1 MMP-9 binding pockets. All compounds fit 1GKC and 2OW1 binding domains as shown in Fig 10A. The overlaying of the induced-fit docked pose of compound **27** with the co-crystalized ligand (NFH) is shown in Fig 10B. Compound **27** shows the highest induced-fit docking score (−11.10 kcal/mol) and forms four H-bonds with PRO421, LEU188, and TYR 393. While the Glide docking of compound **27** gives a −9.09 kcal/mol docking score by forming three H-bonds with TYR420 and GLY 186.

Indeed, the IFD scores for the most synthesized compounds were shown to be more negative than the Glide docking scores. This is because Glide docking assumes that the protein`s structure is rigid and focuses on fitting the ligand into the rigid protein conformation without considering the protein`s flexibility [44]. Accordingly, the loss of optimal binding mode between ligand and protein structure arises from flexible conformational changes upon complex formation [34]. On the other hand, IFD allows the flexibility of both the ligand and protein to optimize their conformations during docking to

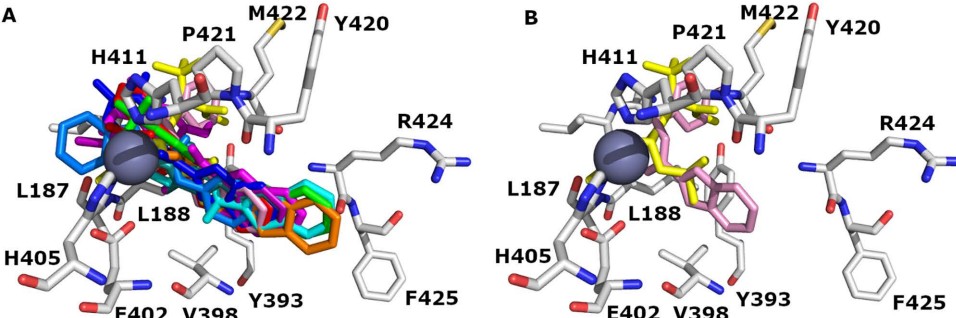

**Fig 10. The binding domains of wild MMP-9 (PDB ID: 1GKC).** (A) Orientation of induced-fit docked poses of 1H-Indole-2-carboxamide compounds and (**B**) overlaying of the induced-fit docked pose of **compound 27** with co-crystalized ligand (NFH) (yellow color). Carbon (C) atoms are depicted in grey colors, N in blue color, and O in red color. **Compound 29** is depicted in marine color, **compound 20** in green color, **compound 21** in red color, **compound 22** in blue color, **compound 23** in magenta color, **compound 24** in cyan color, **compound 25** in orange color, **compound 26** in purple color, **compound 27** in pink color, and **compound 28** in deep teal color. Zinc metal is presented as a sphere (metallic color). H bonds and valence were removed for clarification. Picture visualized by PYMOL [37].

improve the accuracy of the binding prediction [35]. This might reflect the actual behavior of both ligand and protein in nature [36,45].

The purchased hits (**NSC3276** and **NSC123019**) were docked against the WT MMP-9 (1GKC) and MUT MMP-9 (2OW1) binding pocket (Table 2). Both hits fit 1GKC and 2OW1 binding domains as shown in Fig 11. **NSC3276** fit binding pocket scoring −9.71 Kcal/mol through one hydrogen bond with TYR420. Meanwhile, **NSC123019** engages the MMP-9 binding pocket scoring −9.17 Kcal/mol through four H-bonds with GLY 186, LEU188, PRO421, and TYR423.

The superposition of the IF-docked co-crystalized ligand (NFH) pose and its original conformation in 1GKC is shown in Fig 12 [18]. The root mean square deviation (RMSD) for NHF heavy atoms is 0.1530 A°. Similarly, in 2OW1, the

**Table 2. The Glide and induced-fit docking scores (kcal/mol) and interaction forces of purchased partial-match hits against 1GKC and 2OW1.**

| # NSC Number | Code | 1GKC | | 2OW1 | | 1GKC | | 2OW1 | |
| --- | --- | --- | --- | --- | --- | --- | --- | --- | --- |
| | | Glide docking score (kcal/mol) | Interaction Forces with 1GKC | Glide docking score (kcal/mol) | Interaction Forces with 2OW1 | IF-docking score (kcal/mol) | Interaction Forces with 1GKC | IF-docking score (kcal/mol) | Interaction Forces with 2OW1 |
| 3276 | Compound 1 | −8.10 | -One coordination with $Zn^{+2}$ -One π-π stacking with HIS401 -Three halogen bonds with LEU188 and ALA189 -One H-bond with TYR420 | −7.99 | -One coordination with $Zn^{+2}$ -One π-π stacking with HIS401 -Two halogen bonds with LEU188 and ALA189 -One H-bond with TYR420 | −9.71 | -One coordination with $Zn^{+2}$ -One π-π stacking with HIS401 -Two halogen bonds with LEU188 and ALA189 -One H-bond with TYR420 | −9.32 | -One coordination with $Zn^{+2}$ -One π-π stacking with HIS401 -Two halogen bonds with LEU188 and ALA189 -One H-bond with TYR420 |
| 123019 | Compound 2 | −7.36 | -Two coordination with $Zn^{+2}$ -Three H-bonds with MET 422, TYR420, and PRO421 | −7.22 | -Two coordination with $Zn^{+2}$ -Two H-bonds with TYR420, and PRO421 | −9.17 | -Two coordination with $Zn^{+2}$ -Two covalent bonds with $Zn^{+2}$ -Four H-bonds with GLY 186, PRO421, LEU188, and TYR423 | −9.17 | -Two coordination with $Zn^{+2}$ -Two covalent bonds with $Zn^{+2}$ -Three H-bonds with PRO421, LEU188, and TYR423 |

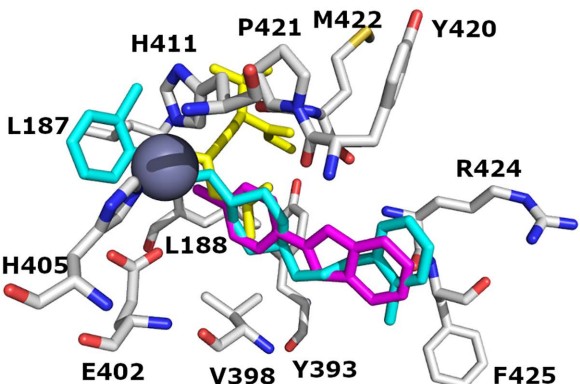

**Fig 11. The orientation of induced-fit docked poses of NSC3276 and NSC123019 with co-crystalized ligand (NFH) (yellow color) in 1GKC.**
Carbon (C) atoms are depicted in grey colors, N in blue color, and O in red color. **NSC3276** is depicted in magenta color and **NSC123019** in cyan color. H bonds and valence were removed for clarification. Zinc metal is presented as a sphere (metallic color). Picture visualized by PYMOL [37].

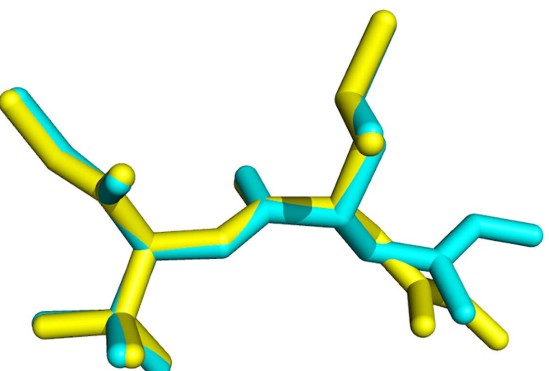

**Fig 12. The superposition of the IF-docked NFH pose (yellow color) and its native conformation (cyan color) in 1GKC [18].** Picture made by PYMOL [37].

superposition of the IF-docked co-crystalized ligand (7MR) and its native conformation was performed and gave an RMSD value equal to 0.2496 A° (Fig 13). The RMSD value indicates the reliability of the IFD program in generating the native pose [16]. The results indicate that IFD can successfully determine the native pose in crystal coordinates and, consequently, predict the ligand-binding pose.

## In-Vitro evaluation

The inhibitory potential of all tested compounds against MMP-9 was evaluated through a complete assay system (Abcam, catalogue # ab139448) in terms of the percentage inhibition at 50 μM in comparison to the reference MMP-9 inhibitor *N*-Isobutyl-*N*-(4-methoxyphenylsulfonyl) glycyl hydroxamic acid (**NNGH**) (Fig 14) (Table 3) through a colorimetric assay

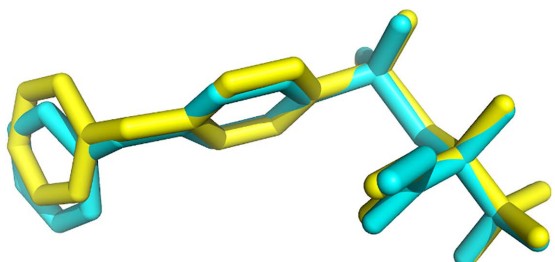

**Fig 13. The superposition of the IF-docked 7MR pose (yellow color) and its native conformation (cyan color) in 2OW1 [16].** Picture made by PYMOL [37].

**Fig 14. Chemical structure of NNGH.**

**Table 3. Inhibitory profiles and 50% inhibitory concentrations (IC$_{50}$) of the tested compounds against MMP-9.**

| No. | Compounds | % Inhibition (50 µM) | IC$_{50}$ (µM) |
|---|---|---|---|
| 1. | NNGH | 95.02* | 47.81** |
| 2. | Compound 1 | 60.24 | 36.77 |
| 3. | Compound 2 | 76.14 | 28.59 |
| 4. | Compound 5 | 16.57 | – |
| 5. | Compound 6 | 3.53 | – |
| 6. | Compound 7 | 59.48 | – |
| 7. | Compound 8 | 61.06 | 39.96 |
| 8. | Compound 9 | 58.01 | – |
| 9. | Compound 10 | 62.94 | 33.29 |
| 10. | Compound 11 | 37.01 | – |
| 11. | Compound 12 | 42.91 | – |
| 12. | Compound 13 | 26.33 | – |
| 13. | Compound 14 | 55.80 | – |
| 14. | Compound 15 | 37.38 | – |
| 15. | Compound 20 | 76.14 | 30.82 |
| 16. | Compound 21 | 64.22 | 34.84 |
| 17. | Compound 22 | 59.48 | – |
| 18. | Compound 23 | 40.37 | – |
| 19. | Compound 24 | 22.65 | – |
| 20. | Compound 25 | 2.39 | – |
| 21. | Compound 26 | 35.54 | – |
| 22. | Compound 27 | 65.93 | 34.84 |
| 23. | Compound 28 | 52.11 | – |
| 24. | Compound 29 | 63.16 | 37.28 |

µM: micromolar; * percent inhibition of NNGH at 1.3 µM, ** nM: nanomolar.

using thiopeptide chromogenic substrate (Ac-PLG-[2-mercapto-4-methyl-pentanoyl]-LG OC$_2$H$_5$). The MMP-9 cleavage site peptide bond is replaced by a thioester bond in the thiopeptide. The cleavage of this bond results in a sulfhydryl group, which reacts with DTNB [5,5'-dithiobis (2-nitrobenzoic acid), Ellman's reagent] to form 2-nitro-5-thiobenzoic acid, which can be easily detected at 412 nm.

In this study, we evaluate the inhibitory effect of the tested compound on MMP-9 enzymatic activity using a colorimetric activity assay kit (ab139448). These inhibitors revealed a significant inhibition of MMP-9 activity compared to untreated controls (P < 0.0001). Eight compounds (**1, 2, 8, 10, 20, 21, 27, and 29**) revealed more than 60% inhibition of the MMP-9 enzyme, which belongs to the benzamide, 1H-indole-2-carboxamide scaffold, and partial-match hits. These findings provide an experimental confirmation of its predicted binding affinity observed in silico through molecular docking approaches. The IF docking tool was used, revealing a strong binding interaction between these active inhibitors and the MMP-9 active site, with docking scores ranging from −8.35 to −11.96 kcal/mol. These docking scores are approximations of binding free energy, with more negative values indicating stronger binding in the active site. Our results correlate well with this prediction, indicating that the active inhibitor not only binds efficiently to the active site but also effectively blocks its proteolytic activity.

In literature, several studies proved the effectiveness of the indole ring in inhibiting MMP-9 activity through the formation of specific hydrogen bonds with GLU 402, TYR420, PRO421, LEU188, and TYR393, which is comparable

to the H-Bonds formed by compounds **20, 21, 27,** and **29** [46–49]. In MMP-9, the active site is composed of the S1` hydrophobic pocket, so placing a trifluoromethyl group near this pocket enhances the topological fit of the inhibitor through van der Waals interactions, improving binding affinities as in compounds **20** and **21** [16,50]. Regarding compound **20**, substituted with a trifluoromethyl group at the *m*-position, showed a higher binding affinity (−10.73 kcal/mol) than compound **21**, which is substituted with an *o*-trifluoromethyl group (−8.35 kcal/mol). The substitution in the *m*-position allowed better accommodation of the ligand inside the S1` pocket with less steric hindrance compared to the substitution in the *o*-position [51]. These results aligned with the active MMP-9 percent inhibition of these two inhibitors (76.14% and 64.22%, respectively). Compound **8** is substituted with a benzothiazole ring, which was previously proven to enhance binding affinity through π-π stacking interactions with HIS401 and PHE110 in the MMP-9 binding pocket [52,53].

This assay specifically detects the active form of MMP-9; the observed decrease in absorbance indicates direct inhibition of MMP-9 proteolytic activity rather than changes in protein expression. This point is critical, as it highlights the importance of targeting and neutralizing MMP-9 activity, which is often upregulated in several diseases such as cancer.

## Conclusion

Our study portrays the design, synthesis, and experimental evaluation of potential MMP-9 inhibitors. We identified the pharmacophoric features of MMP-9 inhibitors and disclosed a potential MMP-9 inhibitor hit targeting the active site. **Compounds 2 and 20** showed more than 70% inhibition of MMP-9, with $IC_{50}$ values of 28.59 μM and 30.82 μM, respectively. The IF docking tool revealed strong binding interactions between these two inhibitors and the MMP-9 active site, with docking scores ranging from −9.17 to −10.73 kcal/mol. The observed alignment between the computational approach and experimental validation strengthens the results about the inhibitor`s specificity and potency. In addition, this validates the docking model and suggests that the predicted binding pose most likely reflects the biologically key interactions.

## Supporting information

**S1 File. Spectroscopic Charts S1-S21.** $^{1}$H-NMR, $^{13}$C-NMR, and HRMS (ESI) spectra of synthesized compounds.
(DOCX)

**S1 Table. Reported selective MMP-9 inhibitors with their $IC_{50}$ values.**
(DOCX)

**S2 Table. The Glide docking score (Kcal/mol) and interaction forces of the full match hits-based on virtual screening.**
(DOCX)

**S3 Table. The Glide docking score (kcal/mol) and interaction forces of the partial match hits-based on virtual screening.**
(DOCX)

**S4 Table. The Glide docking score (kcal/mol) and interaction forces of reported inhibitors against WT MMP-9 (PDB ID: 1GKC) and MUT MMP-9 (PDB ID: 2OW1).**
(DOCX)

**S5 Table. The induced fit docking scores (kcal/mol) and interaction forces of reported inhibitors against 1GKC and 2OW1.**
(DOCX)

## Acknowledgments

The authors acknowledge the Deanship of Scientific Research at the University of Jordan, the School of Pharmacy for providing cell culture laboratories and equipment, and the Department of Chemistry for spectroscopic facilities. The authors acknowledge the Deanship of Scientific Research at Al-Zaytoonah University of Jordan and the Faculty of Pharmacy for providing professional chemical, modeling, and bioinformatics laboratories facilities, as well as computational resources and databases.

## Author contributions

**Conceptualization:** Dima Sabbah, Kamal Sweidan, Rima Hajjo, Sanaa K. Bardaweel.

**Data curation:** Dima Sabbah, Zainab Ahmed Rashid, Rima Hajjo, Sanaa K. Bardaweel.

**Formal analysis:** Dima Sabbah, Zainab Ahmed Rashid, Kamal Sweidan, Rima Hajjo, Sanaa K. Bardaweel.

**Funding acquisition:** Dima Sabbah, Rima Hajjo, Sanaa K. Bardaweel.

**Investigation:** Dima Sabbah, Rima Hajjo, Sanaa K. Bardaweel.

**Methodology:** Dima Sabbah, Zainab Ahmed Rashid, Kamal Sweidan, Rima Hajjo, Sanaa K. Bardaweel.

**Project administration:** Dima Sabbah, Rima Hajjo, Sanaa K. Bardaweel.

**Resources:** Dima Sabbah, Rima Hajjo, Sanaa K. Bardaweel.

**Software:** Dima Sabbah, Rima Hajjo, Sanaa K. Bardaweel.

**Supervision:** Dima Sabbah, Kamal Sweidan, Rima Hajjo, Sanaa K. Bardaweel.

**Validation:** Dima Sabbah, Rima Hajjo, Sanaa K. Bardaweel.

**Visualization:** Dima Sabbah, Zainab Ahmed Rashid, Kamal Sweidan, Rima Hajjo, Sanaa K. Bardaweel.

**Writing – original draft:** Zainab Ahmed Rashid.

**Writing – review & editing:** Dima Sabbah, Kamal Sweidan, Rima Hajjo, Sanaa K. Bardaweel.

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
