## [Decision Letter · Decision Letter 0]

21 Oct 2025

Dear Dr. Sabbah,

Thank you for submitting your manuscript to PLOS ONE. After careful consideration, we feel that it has merit but does not fully meet PLOS ONE’s publication criteria as it currently stands. Therefore, we invite you to submit a revised version of the manuscript that addresses the points raised during the review process.

We look forward to receiving your revised manuscript.

Kind regards,

Ahmed Elkamhawy

Academic Editor

PLOS ONE

Journal Requirements:

2. We note that this submission includes NMR spectroscopy data. We would recommend that you include the following information in your methods section or as Supporting Information files:

a) The make/source of the NMR instrument used in your study, as well as the magnetic field strength. For each individual experiment, please also list: the nucleus being measured; the sample concentration; the solvent in which the sample is dissolved and if solvent signal suppression was used; the reference standard and the temperature.

b) A list of the chemical shifts for all compounds characterised by NMR spectroscopy, specifying, where relevant: the chemical shift (δ), the multiplicity and the coupling constants (in Hz), for the appropriate nuclei used for assignment.

c)The full integrated NMR spectrum, clearly labelled with the compound name and chemical structure.

We also strongly encourage authors to provide primary NMR data files, in particular for new compounds which have not been characterised in the existing literature. Authors should provide the acquisition data, FID files and processing parameters for each experiment, clearly labelled with the compound name and identifier, as well as a structure file for each provided dataset. See our list of recommended repositories here: https://journals.plos.org/plosone/s/recommended-repositories

3. Please note that PLOS One has specific guidelines on code sharing for submissions in which author-generated code underpins the findings in the manuscript. In these cases, we expect all author-generated code to be made available without restrictions upon publication of the work. Please review our guidelines at https://journals.plos.org/plosone/s/materials-and-software-sharing#loc-sharing-code and ensure that your code is shared in a way that follows best practice and facilitates reproducibility and reuse.

4.If the reviewer comments include a recommendation to cite specific previously published works, please review and evaluate these publications to determine whether they are relevant and should be cited. There is no requirement to cite these works unless the editor has indicated otherwise.

Reviewers' comments:

Reviewer's Responses to Questions

**Comments to the Author**

1. Is the manuscript technically sound, and do the data support the conclusions?

Reviewer #1: Yes

Reviewer #2: Yes

2. Has the statistical analysis been performed appropriately and rigorously?

Reviewer #1: Yes

Reviewer #2: N/A

3. Have the authors made all data underlying the findings in their manuscript fully available?

Reviewer #1: Yes

Reviewer #2: Yes

4. Is the manuscript presented in an intelligible fashion and written in standard English?

Reviewer #1: Yes

Reviewer #2: Yes

Reviewer #1: This study identifies potential inhibitors of matrix metalloproteinase-9 (MMP-9), a zinc-dependent enzyme linked to various diseases. Several compounds demonstrated over 60% inhibition, with compounds 2 and 20 showing IC50 values of 28.59 μM and 30.82 μM. Computational docking aligned with experimental validation, confirming specificity, potency, and key biological interactions.

However, there are several shortcomings in this work that need to be addressed. The authors should carefully revise the manuscript for clarity, correct grammatical and typographical errors, ensure consistent formatting, and eliminate all proofreading issues before it can be considered further.

• Many chemical structures are not geometrically accurate and need to be redrawn.

• The amide bonds in all chemical structures should ideally be shown in the anti- conformation, which is more stable than the syn- conformation.

• Abbreviations need to be consistent throughout the manuscript:

• Use either hours, hr, or h for reaction time, not a mix.

• Temperature should be written as “X ºC” (with a space), not “XºC.”

• “Diethyl ether” should be written without a hyphen.

• “Rf” should be written correctly in the experimental details.

• The (M+Na)+ values in compound 8 need correction for significant figures.

• In the general methods, the stoichiometry of amines is written as 1:1 eq. This should be clarified and corrected.

• The use of 6 equivalents of pyridine is unusually large compared to the amine. The authors should justify this choice, as typically a 1:1 ratio of base/amine is sufficient.

• In the synthesis section, the statement “the product was collected after solvent evaporation, followed by recrystallization with CHCl3/n-hexane (19:20)” seems questionable. The solvent ratio should be verified.

• In Scheme 2, the condition “2) 80 °C, 2 hours” is unclear. Please clarify what it refers to.

• In Scheme 2, the condition “CHCl3, pyridine, 80 °C, 24 hr” is problematic since chloroform boils at 61 ºC. The authors should clarify whether this refers to reflux conditions. The same applies to Scheme 1.

• The target compounds are incorrectly referred to as “benzoyl chloride” and “indole-2-carbonyl chloride.” These are amides, not acyl chlorides. The nomenclature must be corrected.

• In Table 6, % inhibition and IC50 values are reported with inconsistent significant figures. These should be standardized.

Reviewer #2: The manuscript PONE-D-25-47277 presents a new approach to developing a pharmacophore model for inhibitors of matrix metalloproteinase-9 (MMP-9). The development of inhibitors of MMP-9 would benefit therapeutic approaches to rheumatoid arthritis, atherosclerosis, and cancer metastasis. The authors have identified several potential lead compounds through computational docking and assayed their inhibitor activity.

The research and approach to this problem are significant advancements in knowledge. The manuscript is recommended for publication after major revisions, particularly to the narrative and figures for the Results and Discussion section.

A minor correction is needed on line 441 for the inhibitor concentration, should be 50 �M.

Figure 4 should be remade to match the proportions of figure 3. The images appear to be stretched and squashed by inappropriate resizing.

The descriptions on line 558 about “lung cancer cell lines” and on line 565 seem inaccurate regarding molecular modeling for the “anticancer effect of MMP-9 inhibitors against lung cancer cell lines.” The phrasing seems unnecessary and is not mentioned elsewhere and the no assays using lung cancer cell lines were included in this study. Only the colorimetric inhibitor screening assay was reported in this manuscript.

The remainder of the Results and Discussion starting on line 588 needs to be carefully rewritten by the authors. Several mistakes were found by this reviewer and there are likely more problems that should be addressed in a major revision of this section.

The paragraph starting on line 588 seems incomplete and does not identify important amino acids that are referenced in the following paragraph. On line 594, reference 65 seems incorrect as note in more detail below since the reference study focused on antibodies and allosteric inhibition of MMP-9. The description of a “another” study starting on line 596 does not include a reference to that study. On line 604, “docking study” starts an incomplete sentence. The reference 65 describes the antigenic epitope residues of Arg162, Glu111, Asp113 and Ile198, but made no mention of any residues near the substrate binding site. Their assays suggest that the antibodies are non-competitive and allosteric inhibitors. On line 609, the experimental crystal structure (1GKC) is incorrectly presented as a docking study.

The results of the molecular docking of synthesized compounds and commercially available compounds is not presented in a way that can be appreciated by the reader (note there are no line numbers in this portion of the manuscript). The description of compound 8 docking into MMP-9 (1GKC) highlights hydrogen bonding interactions with Arg424, Leu188 and Pro421. In figure 6B, compound 8 is not close enough to Arg424 to form a hydrogen bond and neither Leu188 nor Pro421 are shown in the figure. If these interactions are only apparent after Glide of induced fit modeling, a figure of showing the modeled MMP-9 structure should be included. Similarly, for compound 27, there are hydrogen bonds (inconsistently called H-bonds) to Pro421, Leu188, Tyr393 and Tyr420 described in the docking results. In figure 7B, neither Pro421 nor Leu188 are visible and both Tyr393 and Tyr420 appear too far away. Again a figure of the docking model should be shown to highlight these interactions which may not be apparent in the superposition with the crystal structure. Lastly, the docking of the commercially available compounds NSC3276 and NSC123019 both appear to have steric clashes with the backbone of Met 422 in figure 8. Again, this may be an artifact of the superposition and should be addressed by include figures of the docked protein structures.

The supplementary tables S4 and S5 cite references that are not specified in the manuscript bibliography. The bibliography for the supplementary material appears to have been omitted from the submission.

Please note that the figure legends appearing within the narrative appears to be an formatting mistake and detracts from the review process.

**Do you want your identity to be public for this peer review?** For information about this choice, including consent withdrawal, please see our Privacy Policy

Reviewer #1: No

Reviewer #2: No

---

## [Author Response · Author response to Decision Letter 1]

29 Oct 2025

The Rebuttal file embeds the answers to each comments and changes we made.

---

## [Decision Letter · Decision Letter 1]

10 Nov 2025

Integrative computational, synthetic, experimental evaluation of targeted inhibitors against Matrix Metalloproteinase-9: Toward precision modulation of proteolytic activity

PONE-D-25-47277R1

Dear Dr. Sabbah,

We’re pleased to inform you that your manuscript has been judged scientifically suitable for publication and will be formally accepted for publication once it meets all outstanding technical requirements.

Kind regards,

Ahmed Elkamhawy

Academic Editor

PLOS ONE

Additional Editor Comments (optional):

Reviewers' comments:

Reviewer's Responses to Questions

**Comments to the Author**

Reviewer #1: All comments have been addressed

Reviewer #2: All comments have been addressed

2. Is the manuscript technically sound, and do the data support the conclusions?

Reviewer #1: Yes

Reviewer #2: Yes

3. Has the statistical analysis been performed appropriately and rigorously?

Reviewer #1: Yes

Reviewer #2: N/A

4. Have the authors made all data underlying the findings in their manuscript fully available?

Reviewer #1: Yes

Reviewer #2: Yes

5. Is the manuscript presented in an intelligible fashion and written in standard English?

Reviewer #1: Yes

Reviewer #2: Yes

Reviewer #1: (No Response)

Reviewer #2: No preference on arrangement of Figure 4A and 4B. Proportionality has been corrected and looks much better.

**Do you want your identity to be public for this peer review?** For information about this choice, including consent withdrawal, please see our Privacy Policy

Reviewer #1: No

Reviewer #2: No

---

## [Editor Report · Acceptance letter]

PONE-D-25-47277R1

PLOS One

Dear Dr. Sabbah,

I'm pleased to inform you that your manuscript has been deemed suitable for publication in PLOS One. Congratulations! Your manuscript is now being handed over to our production team.

Kind regards,

on behalf of

Dr. Ahmed Elkamhawy

Academic Editor

PLOS One